# Organic phosphorus cycling may control grassland responses to nitrogen deposition: a long-term field manipulation and modelling study

Christopher R. Taylor[1], Victoria Janes-Bassett[3], Gareth Phoenix[1], Ben Keane[1], Iain P. Hartley[2], Jessica A.C. Davies[3]

[1]Department of Animal and Plant Sciences, University of Sheffield, Sheffield, UK

[2]Geography, College of Life and Environmental Science, University of Exeter, Exeter, UK

[3]Lancaster Environment Centre, Lancaster University, Lancaster, UK

Corresponding author: Christopher Taylor (ctaylor8@sheffield.ac.uk)

**Abstract**

Phosphorus (P) limited ecosystems are widespread, yet there is limited understanding of how these ecosystems may respond to anthropogenic deposition of nitrogen (N), and the interconnected effects on the biogeochemical cycling of carbon (C), N and P. Here, we investigate the consequences of enhanced N addition on the C-N-P pools of two P-limited grasslands; one acidic and one limestone, occurring on contrasting soils and explore their responses to a long-term nutrient-manipulation experiment. We do this by combining data with an integrated C-N-P cycling model (N14CP). We explore the role of P-access mechanisms by allowing these to vary in the modelling framework, and comparing model plant-soil C-N-P outputs to empirical data. Combinations of organic P access and inorganic P availability most closely representing empirical data were used to simulate the grasslands and quantify their temporal response to nutrient manipulation. The model suggested that access to organic P is a key determinant of grassland nutrient limitation and responses to experimental N and P manipulation. A high rate of organic P access allowed the acidic grassland to overcome N-induced P limitation, increasing biomass C input to soil and promoting SOC sequestration in response to N addition. Conversely, poor accessibility of organic P for the limestone grassland meant N provision exacerbated P-limitation and reduced biomass input to the soil, reducing soil carbon storage. Plant acquisition of organic P may therefore play an important role in reducing P-limitation, and determining responses to anthropogenic changes in nutrient availability. We conclude that grasslands differing in their access to organic P may respond to N deposition in contrasting ways and where access is limited, soil organic carbon stocks could decline.

## 1. Introduction

Grasslands represent up to a third of terrestrial net primary productivity (NPP) [Hoekstra *et al.* 2005] and potentially hold over 10% of the total organic carbon stored within the biosphere [Jones and Donnelly, 2004]. The ecosystem services provided by grasslands, such as carbon storage, are highly sensitive to perturbations in their nutrient cycling, including the perturbation of nitrogen (N) inputs from atmospheric deposition [Phoenix *et al.* 2012].

Since the onset of the industrial revolution, human activity has doubled the global cycling of N, with anthropogenic sources contributing 210 Tg of fixed N per year to the global N cycle, surpassing naturally fixed N by 7 Tg N yr$^{-1}$ [Fowler *et al.* 2013]. Much of this additional N is deposited on terrestrial ecosystems from atmospheric sources. This magnitude of N deposition results in a range of negative impacts on ecosystems (including grasslands) such as reductions in biodiversity [Bobbink *et al.* 2010; Southon *et al.* 2013], acidification of soil, and the mobilisation of potentially toxic metals [Carroll *et al.* 2003; Horswill *et al.* 2008; Phoenix *et al.* 2012]

Despite large anthropogenic fluxes of N, most terrestrial ecosystems on temperate post-glacial soils are thought to be N-limited (biomass production is most restricted by N availability) [Vitousek and Howarth, 1991; Du *et al.* 2020], as weatherable sources of phosphorus (P) remain sufficiently large to meet plant P demand [Vitousek and Farrington, 1997; Menge *et al.* 2012]. Both empirical and modelling studies have shown that pollutant N, when deposited on N-limited ecosystems, can increase productivity [Tipping *et al.* 2019] and soil organic carbon (SOC) storage [Tipping *et al.* 2017], largely as a result of stimulated plant growth. This suggests that while there are negative consequences of N deposition, there may also be benefits from enhanced plant productivity and increases in carbon sequestration.

Whilst most research focuses on N-limited ecosystems [LeBauer and Treseder, 2008], a number of studies have highlighted that P limitation and N-P co-limitation are just as prevalent, if not more

widespread, than N limitation [Fay *et al.* 2015; Du *et al.* 2020; Hou *et al.* 2020]. In a meta-analysis of
grassland nutrient addition experiments spanning five continents, Fay *et al*. [2015] found that
aboveground annual net primary productivity was limited by nutrients in 31 out of 42 sites, most
commonly through co-limitation of N and P [Fay *et al.* 2015]. Similarly, P additions in 652 field
experiments increased aboveground plant productivity by an average of 34.9% [Hou *et al.* 2020], and
it is estimated that P limitation, alone or through co-limitation with N, could constrain up to 82% of
the natural terrestrial surface's productivity [Du *et al.* 2020].
Furthermore, P limitation may be exacerbated by N deposition [Johnson *et al.* 1999; Phoenix *et al.*
2004], or become increasingly prevalent as previously N-limited ecosystems transition to N-sufficient
states [Goll *et al.* 2012].  For example, in parts of the Peak District National Park, UK, N deposition has
exceeded 3 g m$^{-2}$ yr$^{-1}$, with further experimental additions of 3.5 g m$^{-2}$ yr$^{-1}$ leading to decreases rather
than increases in productivity of limestone grasslands [Carroll *et al.* 2003]. This makes P limitation
critical to understand in the context of global carbon and nutrient cycles. By definition, N deposition
should impact P-limited ecosystems differently to N-limited ones, yet there is little understanding of
how N deposition impacts these systems.
While N deposition may worsen P limitation in some instances, plant strategies for P acquisition may
require substantial investments of N, suggesting that increased N supply may facilitate enhanced P
uptake [Vance *et al.* 2003; Long *et al.* 2016; Chen *et al.* 2020]. Indeed, previous work from long-term
experimental grasslands has shown strong effects of N deposition on plant enzyme production
[Johnson *et al.* 1999; Phoenix *et al.* 2004], whereby the production of additional extracellular
phosphatase enzymes was stimulated. While it is not clear if this response is driven by exacerbated P-
limitation resulting from N deposition or extra N availability making elevated enzyme production
possible, such changes in plant physiology may promote cleaving of P from organic soil pools. Over
time, the accumulation of plant-available P from organic sources may provide a mechanism by which
plants exposed to high levels of N deposition may overcome P limitation [Chen *et al.* 2020].
By using the integrated C-N-P cycle model N14CP, Janes-Bassett *et al.* [2020] suggest that the role of
organic P cycling in models may be poorly represented, as the model failed to simulate empirical yield
data in agricultural soils with low P fertiliser input. Organic P access is therefore likely an important
means of nutrient acquisition for plants in high N and low P soils [Chen *et al.* 2020], yet our
understanding of organic P cycling in semi-natural ecosystems is fairly limited [Janes-Bassett *et al.*
2020]. Such interdependencies of the C, N and P cycles make understanding an ecosystem's response
to perturbations in any one nutrient cycle challenging, particularly when ecosystems are not solely
limited in N. This highlights the need for integrated understanding of plant-soil nutrient cycling across
the C, N and P cycles, and in ecosystems that are not solely N-limited.
Process-based models have a role to play in addressing this, as they allow us to test our mechanistic
understanding and decouple the effects of multiple drivers. There has been increasing interest in
linking C with N and P cycles in terrestrial ecosystem models [Wang *et al.* 2010; Achat *et al.* 2016; Jiang
*et al.* 2019] as the magnitude of the effects that anthropogenic nutrient change can have on
biogeochemical cycling are realised [Yuan *et al.* 2018]. Yet, few modelling studies have explicitly
examined the effects of P limitation, or the role of organic P access in determining nutrient limitation,
likely mirroring the relatively fewer empirical studies of these systems.
By combining process-based models with empirical data from long-term nutrient-manipulation
experiments, we may simultaneously improve our understanding of empirical nutrient limitation, the
role(s) of organic P acquisition, and their interactions with anthropogenic nutrient pollution. In
particular, this approach offers a valuable opportunity for understanding ecosystem responses to
environmental changes that may only manifest after extended periods of time, such as with changes
in soil organic C, N and P pools, which typically occur on decadal timescales [Davies *et al.* 2016a, Janes-
Bassett *et al.* 2020]. Here, we combine new data from a long-term nutrient manipulation experiment
on two P-limited upland grasslands (acidic and limestone) occurring on contrasting soils, with the
mechanistic C-N-P plant-soil biogeochemical model; N14CP [Davies *et al.* 2016b].
We use these experimental data to explore the role of organic P access in determining ecosystem
nutrient limitation and grassland responses to long-term nutrient manipulations. Specifically, we aim
to explore how variation in P acquisition parameters, that control access to organic and inorganic
sources of P in the model, may help account for differing responses of empirical grassland C, N and P
pools to N and P additions. Second, we explore the effects of long-term anthropogenic N deposition
and experimental N and P additions on plant and soil variables of the simulated acidic and limestone
grasslands. This will help improve our understanding of organic P process attribution within the model
and may suggest how similarly nutrient limited grasslands could respond to similar conditions.
We hypothesise that 1) access to organic P will be an important determinant of ecosystem nutrient
limitation, 2) increased organic P availability may alleviate P limitation resulting from N deposition and
3) grasslands capable of accessing sufficient P from organic forms may overcome P limitation resulting
from N deposition and nutrient treatments, whereas grasslands lacking such accessibility will not.



**2. Methods**
**2.1. Field experiment description**
The empirical data is from Wardlow Hay Cop (henceforth referred to as Wardlow), a long-term
experimental grassland site in the Peak District National Park (UK) [Morecroft *et al.* 1994]. Details of
empirical data collection are available in supplementary section 1.  There are two distinct grassland
communities occurring in close proximity; acidic (National vegetation classification U4e) and
limestone (NVC CG2d) semi-natural grasslands (Table S2). Both grasslands share a carboniferous
limestone hill but the limestone grassland sits atop a thin humic ranker [Horswill *et al.* 2008] and
occurs predominantly on the hill brow. In contrast, the acidic grassland occurs in the trough of the
hill, allowing the accumulation of wind-blown loess and the formation of a deeper soil profile of a
palaeo-argillic brown earth [Horswill *et al.* 2008].
Despite contrasting soil types, both the acidic and limestone grasslands are largely P-limited
[Morecroft *et al.* 1994; Carroll *et al.* 2003], though occasional N and P co-limitation can occur
[Phoenix *et al.* 2003] and more recently, positive growth responses in solely N-treated plots have
been observed, in line with the latest understanding that long-term N loading may increase P supply
by increasing phosphatase enzyme activity [Johnson et al. 1999; Phoenix et al.2004; Chen et al.
2020].
Nutrients (N and P) have been experimentally added to investigate the effects of elevated N
deposition and the influence of P limitation [Morecroft *et al.* 1994]. Nitrogen treatments simulate
additional N deposition to the background level and the P treatment acts to alleviate P limitation.
Nutrients are added as solutions of distilled water and applied as fine spray by backpack sprayer, and
have been applied monthly since 1995, and since 2017 bi-monthly. Nutrient additions are in the
form of $NH_4NO_3$ for nitrogen and $NaH_2PO_4.H_2O$ for phosphorus. Nitrogen is applied at rates of 0
(distilled water control – 0N), 3.5 (low nitrogen – LN) and 14 g N $m^{-2}$ $yr^{-1}$ (high nitrogen – HN). The P
treatment is applied at a rate of 3.5 g P $m^{-2}$ $yr^{-1}$ (phosphorus – P).
Data collected from the Wardlow grasslands for the purpose of this work are; aboveground biomass
C, SOC, and total N, which is assumed to be equivalent to modelled SON. This new data is combined
with total P data that was collected by Horswill *et al*. at the site [Horswill *et al.* 2008]. Summaries of
these data are available within the supplementary material (Table S1) and details of their collection
and conversion to model-compatible units in supplementary section 1.

**2.2. Summary of model processes**
2.2.1. N14CP model summary
The N14CP ecosystem model is an integrated C-N-P biogeochemical cycle model that simulates net
primary productivity (NPP), C, N and P flows and stocks between and within plant biomass and soils,
and their associated fluxes to the atmosphere and leachates [Davies *et al.* 2016b]. N14CP was
originally developed and tested on 88 northern Europe plot-scale studies, including grasslands,
where C, N and P data were available. All but one of the tested ecosystems exhibited N limitation
[Davies *et al.* 2016b]. It has also been extensively and successfully blind-tested against SOC [Tipping
*et al.* 2017] and NPP data from unimproved grassland sites across the UK [Tipping *et al.* 2019].
However, N14CP has not been extensively tested against sites known to exhibit P limitation,
especially where these are explicitly manipulated by long term experimental treatments. While the
importance of modelled weatherable P ($P_{Weath0}$) and historic N deposition on N-limited C, N and P
have been investigated [Davies *et al.* 2016b], the potential influence of organic P on ecosystem
nutrient limitation and responses to nutrient perturbations have yet to be explored.
Here, we modify N14CP to add experimental N and P additions to simulate a long-term nutrient
manipulation experiment similar to that at the limestone and acidic grasslands at Wardlow, and we
use empirical data from Wardlow to explore the role of organic P cleaving in determining ecosystem
state.  A full model description can be found in Davies *et al.* [2016b], however, a summary of the
most relevant features is given here for convenience.
2.2.2. Net primary productivity and nutrient limitations
Plant biomass is simulated in the model as two sets of pools of coarse and fine tissues representing
both above and belowground plant C, N and P, with belowground biomass for each plant functional
type represented by a root fraction. NPP adds to these on a quarterly basis with growth occurring in
quarters 2 and 3 (spring and summer). In N14CP, NPP depends on a single limiting factor, in
accordance with Liebig's law of the minimum. The factors that can limit growth in the model include
available N and P, temperature or precipitation, the latter two being provided as input driver data
(see section 2.3.2).
First, the potential maximum NPP limited by climate is calculated using regression techniques, as in
Tipping *et al.* [2014]. The corresponding plant demand for N and P to achieve this potential NPP is
then calculated [Davies *et al.* 2016b; Tipping *et al.* 2017]. This demand is defined by plant functional
type stoichiometry, which changes through time in accordance with ecosystem succession (see
section 2.3.2). Stoichiometry of coarse tissue is constant but the fine tissue of each plant functional
type has two stoichiometric end members. This allows the model to represent transitions from N-
poor to N-rich plant communities or an enrichment of the fine tissues within plants (or a
combination of both) [Davies *et al.* 2016b], dependent on available N. This allows a degree of
flexibility in plant C:N ratios in response to environmental changes such as N deposition. If the
available nutrients cannot meet the calculated plant nutrient demand, the minimum calculated NPP
based on either N or P availability is used, giving an estimation of the most limiting nutrient to plant
growth.
Nutrient co-limiting behaviour can occur in the model through increased access to organic P sources
in the presence of sufficient N (see 2.2.3), and by having the rate of N fixation dependent on plant
and microbial available P [Davies *et al.* 2016b]. The initial rate of N fixation is based on literature
values for a given plant functional type and is downregulated by anthropogenic N deposition, but
not soil N content more generally, as it is assumed that atmospherically deposited N is readily
available to N-fixers. Nitrogen fixation in the model is also related to P availability. The degree to
which P availabilty limits this maximum rate of fixation is determined by a constant; $K_{Nfix}$ [Davies et
al. 2016b]. This means that while modelled NPP is limited by availability of a single nutrient, co-
limitation may occur through P limitation of N fixation [Danger *et al.* 2008].

2.2.3. Plant and soil N and P cycling
A simplified summary of key pools and processes regarding plant-soil nutrient cycling are detailed in
Figure 1. Details such as initial base cation pools, their effects on soil pH, and most parameter names
have been omitted for clarity but are available from the original model development study [Davies *et*
*al.* 2016b]. Key changes for the purpose of this work are highlighted in red.
Plant available N is derived from biological fixation, the decomposition of coarse litter and SOM,
atmospheric deposition and direct N application. Fine plant litter enters the SOM pool directly due
to its rapid rate of turnover whereas coarse litter contributes N and P through decomposition and
does not join the SOM pool. Plant available P also comes from SOM and coarse litter decomposition,
direct treatment, desorption of inorganic P from soil surfaces, and sometimes cleaving of organic P
[Davies *et al.* 2016b].  The sorbed inorganic P pool builds over time with inputs of weathered P and
sorption of any excess plant available inorganic P, and desorption occurs as a first order process.
Phosphorus enters the plant-soil system by weathering of parent material, the initial value of which
($P_{Weath0}$ within the model) can be set to a default value, or made site-specific by calibrating this initial
condition to soil observational data (as in methods section 2.3.3). From this initial pool, annual
releases of weathered P are determined by first-order rate constants that are temperature
dependent, with the assumption that no weathering occurs below 0 degrees Celsius. This weathered
P can then contribute toward plant-available P in soil water or be sorbed to soil surfaces. In principle,
P can be added in small quantities by atmospheric deposition [Ridame and Guieu, 2002] but for the
purpose of this work, P deposition is set to zero in the model. While the contribution of P through
atmospheric deposition is increasingly realised [Aciego et al. 2017], we cannot account for the losses
of P that may also occur through landscape redistribution [Tipping et al. 2014].
The size of the available P pool is determined by summing: P retained within plant biomass prior to
litterfall, inorganic P from decomposition, dissolved organic P and P cleaved from SOP by plants.
Accessibility of each P form is determined by a hierarchal relationship in the order mentioned above,
whereby plants and microbes access the most readily available P sources first and only move onto
the next once it has been exhausted.
When N is in sufficient supply and more bioavailable P forms have been exhausted from the total
available pool, simulated plants can access P from SOM via an implicit representation of extracellular
P-cleaving enzymes with a parameter termed $P_{Cleave}$. While empirical data quantifying this parameter
is scarce, N14CP constrains $P_{Cleave}$ by utilising a maximum SOM C:P ratio; $[C:P]_{fixlim}$, that ensures SOM
stoichiometry is not unrealistically disrupted by excessive removal of organic P (Equation 1).

$$P_{Cleave} = SOP - \frac{SOC}{[C:P]_{fixlim}}$$
         Equation 1


The functioning of the $P_{Cleave}$ parameter, including its stoichiometric constraint, remains the same in
this work but we have introduced a modifier to adjust the rate at which plants can access this P
source. This parameter; $P_{CleaveMax}$, represents the maximum amount (g m$^{-2}$ season$^{-1}$) of cleaved P that
plants can acquire from the available P pool to satiate P demand.
A fraction of plant biomass is converted to litter in each quarterly time step and contributes a
proportion of its C, N and P content to SOM, which is sectioned intro three pools (fast, slow and
passive) depending on turnover rate [Davies *et al.* 2016b]. Soil organic P (SOP) is simulated alongside
SOC and SON using C:N:P stoichiometries of coarse and fine plant biomass. Decomposition of SOP,
and its contribution to the available P pool, is subject to the same turnover rate constants as for SOC
and SON.
Carbon is lost as $CO_2$ following temperature-dependent decomposition and as dissolved organic
carbon. Likewise, N and P are lost via dissolved organic N and P in a proportion consistent with the
stoichiometry of each SOM pool. Inorganic N is lost via denitrification and inorganic P can be sorbed
by soil surfaces. Both inorganic N and P can be leached in dissolved forms if they are in excess of
plant demand.


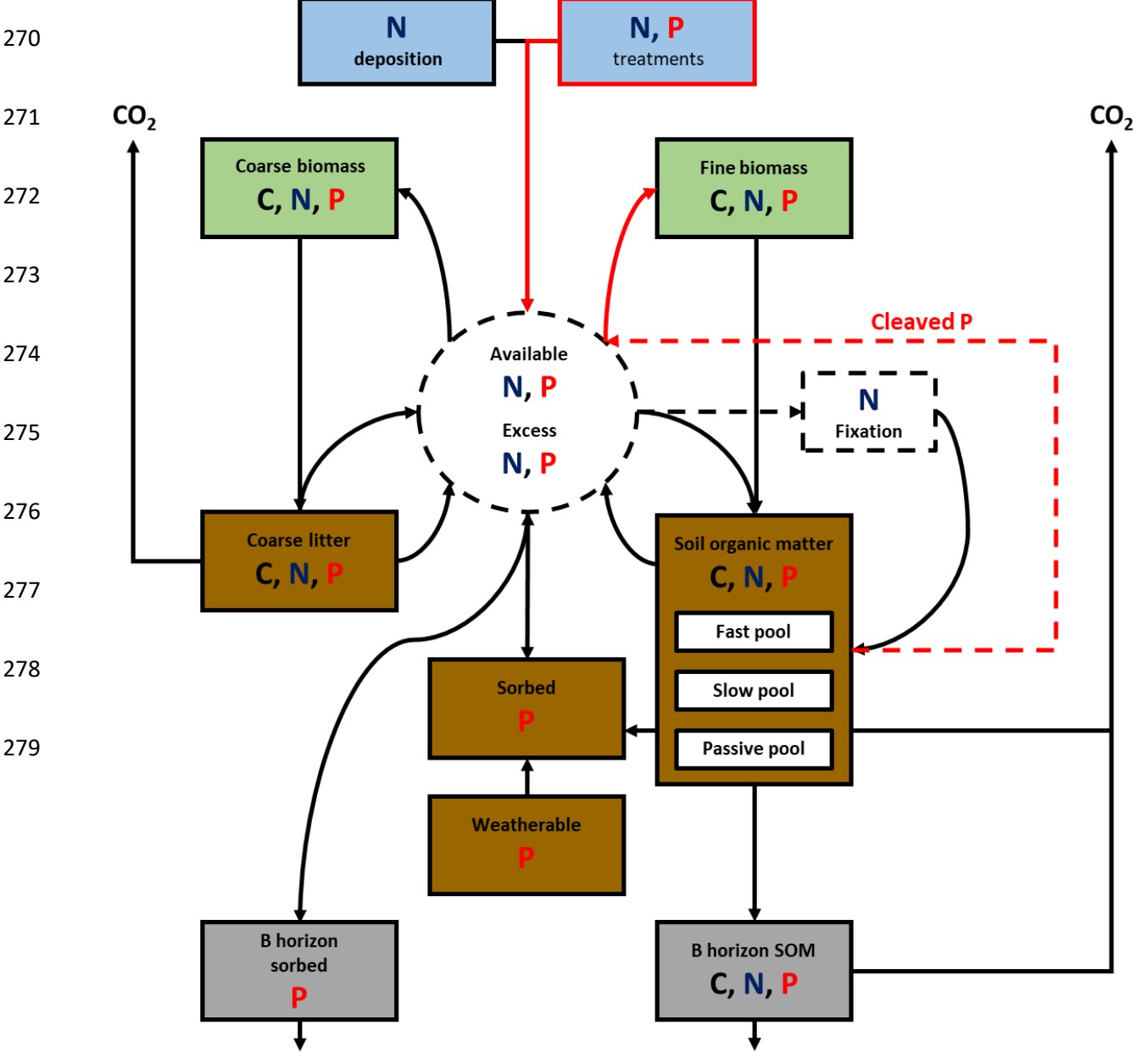

**Figure 1:** A simplified schematic of the key flows and pools of C, N and P within N14CP, adapted from the full schematic available in Davies *et al.* (2016b). Red lines highlight modifications to N14CP for the purpose of this work, including adding experimental nutrients and allowing uptake of cleaved P to be more flexible. Solid lines indicate input to another pool and a dashed line indicates either a feedback or interaction with another pool. In the model, N can enter the available pool via atmospheric deposition, nutrient treatments, biological fixation, and decomposition of coarse litter and SOM. For P, the two main sources are the inorganic sorbed pool and from the turnover of SOM. The former is derived initially from the weatherable supply of P, defined by its initial condition ($P_{Weath0}$). P can also be added to this pool experimentally as with N. The dashed line going from available N and P to N fixation represents the downregulation of N fixation by N deposition and the dependency of N fixation on P availability. The cleaving of organic P from SOM and its incorporation into the plant-available nutrient pool, is represented by the dashed red line and its uptake by plants, determined by $P_{CleaveMax,}$ shown with a solid red line.

**2.3. Simulating the field manipulation experiment with the model**

We use data from the Wardlow limestone and acidic grasslands to explore the potential role organic

P access may have in determining grassland nutrient limitation when exposed to long-term N

deposition and more recently, experimental nutrient manipulation. We use environmental input

data collated from Wardlow to drive model processes. Empirical data regarding contemporary soil C,

N and P for the contrasting grasslands are used to calibrate the initial size of the weatherable P pool

within the model, and to allow access to organic cleaved P to vary to account for patterns in the

data. We do not aim to perfectly replicate the Wardlow grasslands but rather use the unique

opportunity that Wardlow provides to test our understanding of such P-limited ecosystems and how

our conceptualisation of P access mechanisms within the model may affect them. In addition, we

can use the model-simulated grasslands to investigate the potential effects of long-term N

deposition and nutrient manipulation on ecosystems which may differ in their relative availability of

different P forms.

2.3.1. Nutrient applications

Nutrient treatments are treated in N14CP as individual plots in the simulations with differing

amounts of inorganic N and P applied in line with the field experimental treatments (section 2.1).

The N and P treatments are added to the bioavailable N and P pools of the model on a quarterly

basis in line with the model's time-step. While Wardlow nutrient treatments are applied monthly

and N14CP quarterly, the annual sum of applied N or P is equivalent, and nutrients are applied

during all quarters.

2.3.2. Input drivers
N14CP simulations run on a quarterly time step and are spun up from the onset of the Holocene
(10,000 BP in the model). This is to capture the length of time required for soil formation following
deglaciation in north west Europe and is not an attempt to truly model this long term period.
Instead, it allows us to form initial conditions for modern day simulations that takes in what we
know about the site's history and forcings.
To use this spin up phase and simulate contemporary soil C, N and P stocks, we use a variety of input
driver data. Inputs nearer the present are more accurately defined based on site-scale
measurements and assumptions are made regarding past conditions. This approach of spinning up
to present-day observations avoids the assumption that ecosystems are in a state of equilibrium,
which is likely inaccurate for ecosystems exposed to long-term anthropogenic changes in C, N and P
availability. Input driver data include plant functional type history, climatic data and N deposition
data. A summary of the data used for model input is provided in supplementary Table S3. To
simulate the sites' plant functional type history, we used data on Holocene pollen stratigraphy of the
White Peak region of Derbyshire [Taylor *et al.* 1994], which captures important information
regarding Wardlow's land-use history for the entire duration of the model spin up phase.
Input drivers are provided as annual time series to drive the model and as the acidic and limestone
sites are co-located, these input timeseries are shared for both grasslands. It is assumed in the
model that anthropogenic N deposition was negligible prior to 1800 and the onset of the industrial
revolution. After 1800, N deposition is assumed to have increased similarly across Europe [Schopp *et*
*al.* 2003]. In N14CP, this trend is linearly extrapolated from the first year of data (1880) back to 1800
[Tipping *et al.* 2012]. Data regarding N deposition that is specific to Wardlow was incorporated
between the years 2004 and 2014 and the Schöpp *et al.* [2003] anomaly scaled to represent the high
N deposition of the site.
To provide climate forcing data, daily minimum, mean and maximum temperature and mean
precipitation records beginning in 1960 were extracted from the UKPC09 Met office CEDA database
(Table S3). The data nearest to Wardlow was calculated by triangulating latitude and longitude data
and using Pythagoras' theorem to determine the shortest distance. These data were converted into
mean quarterly temperature and precipitation. Prior to this, temperature was assumed to follow
trends described in Davies *et al*. [2016b] and mean quarterly precipitation was derived from Met
Office rainfall data between 1960 to 2016 and held constant.

2.3.3. Model parameters for the acidic and limestone grasslands
The N14CP model has been previously calibrated and tested against a wide range of site data to
provide a general parameter set that is applicable to temperate semi-natural ecosystems, without
extensive site-specific calibration [Davies *et al.* 2016b]. The majority of those parameters are used
here for both grasslands. However, two parameters relating to P sources and processes were
allowed to vary between the sites: the initial condition for the weatherable P pool, $P_{Weath0}$; and the
rate of plant access to organic P sources, $P_{CleaveMax}$ (Figure 1). We allowed $P_{Weath0}$ to vary for each
grassland as variation in a number of factors including lithology and topography mean that we
should expect the flux of weathered P entering the plant-soil system to vary on a site-by-site basis
[Davies *et al.* 2016b]. Indeed, we should expect that $P_{Weath0}$ differs between the acid and limestone
grasslands, as despite their proximity, they have differing lithology.  Davies *et al.* [2016b], show that
variation in this initial condition considerably helps explain variance in contemporary SOC, SON and
SOP stocks between sites. However, it is difficult to set this parameter directly using empirical data,
as information on lithology and P release is limited at the site scale.
As this is the first time that N14CP has been knowingly applied to ecosystems of a largely P-limited
nature, we also allowed the maximum rate at which plants could access cleaved P ($P_{CleaveMax}$) to vary,
to investigate how plant P acquisition might change when more readily accessible P forms become
scarcer. Empirical quantification of organic P access is poor [Janes-Bassett *et al.* 2020], hence we use
a similar data-driven calibration for $P_{CleaveMax}$ as we do for $P_{Weath0}$.
We ran a series of simulations systematically varying $P_{Weath0}$ and $P_{CleaveMax}$ and comparing the results
to observations. We simulated the two grasslands and their treatment blocks with a set of 200
parameter combinations. This captured all combinations of 20 values of $P_{Weath0}$ between 50 and 1000
g m$^{-2}$ and 10 values of $P_{CleaveMax}$ between 0 to 1 g m$^{-2}$ per growing season using a log$_{10}$ spacing to focus
on the lower range of $P_{CleaveMax}$ values. The $P_{Weath0}$ range was set to capture the lower end of $P_{Weath0}$
estimates described in Davies *et al.* [2016b], which were more likely to be appropriate for these P-
poor sites. We explored a range of values for $P_{CleaveMax}$, from zero where no access to organic sources
is allowed, to 1 g m$^{-2}$ per growing season – a rate in the order of magnitude of a fertilizer application.
The model outputs were compared to measured, SOC, SON and total P (Table S4) for each grassland.
We tested how these parameter sets performed by calculating the error between the observations
and model outputs of the same variables for each combination of $P_{CleaveMax}$ and $P_{Weath0}$. The sum of
the absolute errors between modelled and observed soil C, N and P data were scaled (to account for
differing numbers of observations) and summed to provide an F value (Equation 2) as an overall
measure of error across multiple observation variables.


$$F = (\frac{SAE[C_{SOM}]}{\bar{C}_{SOM,Obs}})/C_n + (\frac{SAE[N_{SOM}]}{\bar{N}_{SOM,Obs}})/N_n + (\frac{SAE[P_{Total}]}{\bar{P}_{Total,Obs}})/P_n \qquad \text{(Equation 2)}$$




Plant biomass C data were excluded from the cost function to allow for blind testing of the model's
performance against empirical observations. As the variable most responsive to nutrient additions,
both in terms of rapidity and magnitude of the response, we deemed these the most rigorous data
to use for separate testing. We included soil C, N and P data from all nutrient treatments rather than
just the control to ensure that the selected parameter combination could better account for
patterns in empirical data. For instance, we know that empirical N treatments can increase plant and
soil enzyme activity in both Wardlow grasslands, [Johnson *et al.* 1999; Phoenix *et al.* 2004; Keane *et*
*al.* 2020] which a calibration to control-only data may not have captured.
While the cost function is a useful tool in allowing the model to simulate the magnitude of
contemporary C, N and P pools, it does not allow us to capture all necessary information to
accurately simulate grassland responses to long-term nutrient manipulation. The pattern of
grassland response, i.e. how a variable responds to nutrient treatment, is an important
consideration and is determined in the model by the most limiting nutrient. Consequently, the
parameter combination with the lowest F value, that still maintained a grassland's empirical
response to nutrient additions, was used within the analysis.


**3. Results**

Below, we first present data regarding the results of the calibration of $P_{Weath0}$ and $P_{CleaveMax}$ for each grassland, and how simulated grassland C, N and P using these parameter combinations compares to the empirical data (section 3.1, Figure 2). Raw empirical data is available in table S1 in section 2 of the supplementary material. Second, we explore how the limiting nutrient of the modelled grasslands has changed through time in response to N deposition and experimental treatment (section 3.2, Figure 3). Third, we explore how C, N and P pools in the simulated grasslands have responded to N deposition and nutrient treatment within the model, and include empirical data to contextualise changes (section 3.3, Figure 4). Finally, we present the C, N and P budgets for both modelled grasslands to examine changes in C, N and P pools more closely, in order to better our mechanistic understanding of changes in nutrient flows within the model (section 3.3, Figure 5).

**3.1. Varying phosphorus source parameters**

The model calibration selected parameter values for $P_{Weath0}$ and $P_{CleaveMax}$ that indicate contrasting use of P sources by the two simulated grasslands, with the acidic grassland capable of acquiring more P from organic sources, having a $P_{CleaveMax}$ value of 0.32 g m$^{-2}$ season$^{-1}$ compared to the limestone, with a value 10 times smaller at 0.03 g m$^{-2}$ season$^{-1}$. Conversely, inorganic P availability was greater in the limestone grassland due to the larger weatherable pool of P, $P_{Weath0}$, at 300 g m$^{-2}$ compared to 150 g m$^{-2}$ in the acidic.

The selected parameter combinations resulted in the model simulating the acidic grassland as N-limited and the limestone as P-limited, with reasonable congruence between observed and modelled data. The outputs for the calibrated model are shown in Figure 2 against the observations for above-ground biomass C, soil organic C, and N for both the acidic and limestone grasslands (Fig 2). Raw data used for Figure 2 are provided in supplementary tables S4 and S5.

Overall, N14CP more accurately simulated the magnitude of limestone grassland C, N and P pools
than the acidic, and it generally captured the pattern of responses to nutrient treatment, albeit this
is not always supported by high $r^2$ values. The model estimates of above ground biomass C are
broadly aligned with the observations: capturing variation between the grasslands and treatments
($r^2$ = 0.58), and on average overestimating the magnitude by 12.9% (SE ± 11.9) and 12.1% (SE ± 9.4)
for the acidic and limestone grasslands respectively (Fig 2a).
Soil organic C on average was slightly overestimated (7.1% with SE ± 3.3) for the limestone grassland
(Fig 2b), with a larger average overestimate for the acidic grassland (39.9% with SE ± 6.8). However,
in this latter case the variation between treatments was better captured. Despite a low $r^2$ value for
SOC (0.01), the model broadly captured the patterns we observe in the empirical data, with N
addition increasing SOC in the acidic and P addition increasing SOC in the limestone. However, the
intermediate increase in SOC with P in the acidic grassland is not captured by the model, nor is the
magnitude of the negative effect of LN treatment on limestone SOC.
Simulated magnitudes of SON are well-aligned with observations for the acidic grassland, with an
average error of 2.3% (SE ± 3.2), whilst SON for the limestone grassland was on average
underestimated by 17.8% (SE ± 3.6) (Fig 2c). The variation between treatments was better captured
for acidic than limestone SON but was overall reasonable ($r^2$ = 0.39).
Finally, the model overestimated total soil P (defined in the model as organic P plus sorbed P) by an
average of 6.0% (SE ± 4.3) for the limestone but underestimated by 54.7% (SE ± 8.0) in the acidic
grassland, which was the least accurately predicted variable out of those investigated (Fig 2d). With
only two empirical data points for TP across only two nutrient treatments, it is difficult to discern the
relationship between treatments and TP so an $r^2$ value is of little relevance here.


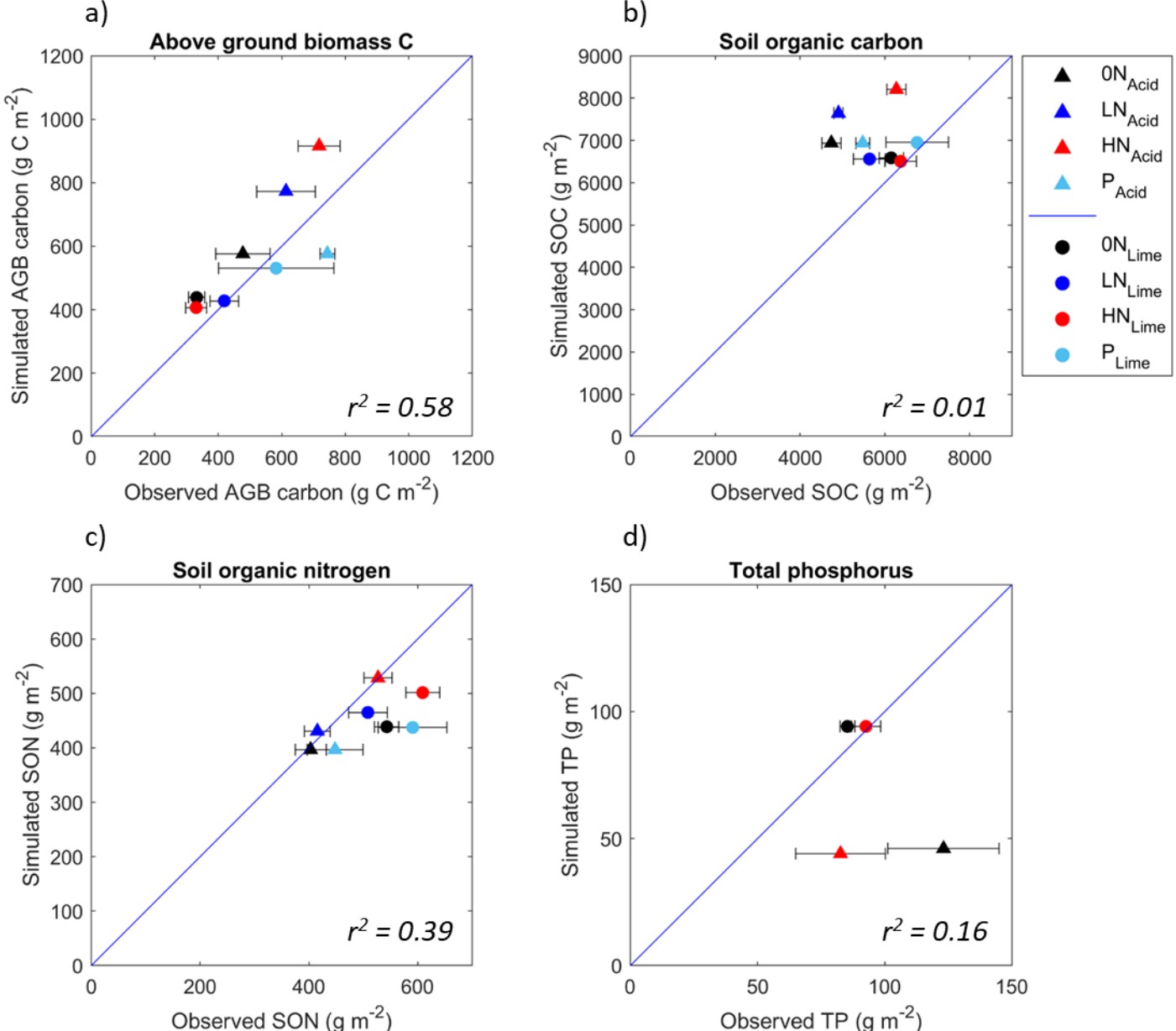

**Figure 2**: A comparison of the observed values of a) aboveground biomass carbon, b) soil organic carbon, c) soil organic nitrogen and d) total soil phosphorus from both grasslands, with simulated values from the model. The blue line represents a 1 to 1 relationship and the closer the data points are to the line, the smaller the discrepancy between observed and modelled data. All data are in grams per metre squared and all treatments for which data were collected are presented. The horizontal error bars represent the standard error of the empirical data means. The $r^2$ value of regression models fitted to the data give an overall indication of the direction of response of each variable to nutrient addition, hence a low value is not necessarily indicative of poor model fit

## 3.2. The limiting nutrient through time

Modelled acid grassland NPP remained N-limited from 1800 through to 2020 under most nutrient treatments (Fig 3). Nitrogen deposition increased the potential NPP through time and the grassland moved toward co-limitation in the LN treatment (i.e. the N and P lines were closer) but remained N-limited (Fig 3b). In the HN treatment, the acidic grassland shifted to P limitation as N-limited NPP surpasses P-limited NPP (Fig 3c).

The simulated limestone grassland was also initially N-limited, but was driven through a prolonged (c. 100 year) state of apparent co-limitation until clearly reaching P-limitation in 1950, solely as a result of N deposition (Fig 3). In the 0N treatment, the grassland remained P-limited but the potential NPP values for N and P are similar, suggesting the grassland is close to co-limitation (Fig 3e). The LN and HN treatment amplified pre-existing P-limitation, lowering the potential NPP of the grasslands (Fig 3f, g). With the addition of P in 1995, P limitation is alleviated, and the ecosystem transitions to a more productive N-limited grassland (Figure 3h).

Another way to interpret the extent of nutrient limitation within N14CP with specific reference to P-demand, is to assess the rate of P cleaving through time. These data corroborate the N and P-limited NPP data, showing that in the limestone grassland, the maximum amount of cleavable P is accessed by plants in the 0N, LN and HN treatments from approximately 1900 through to the end of the experimental period in 2020 (Fig S1, Table S13), highlighting its consistent state of P limitation.

Conversely, while cleaved P is used in the 0N treatment in the acidic grassland, it occurs at approximately one third of the total rate, hence the grassland is not entirely P-limited (Fig S1, Table S9). The LN treatment increases the rate of access to cleaved P and HN causes it to reach its maximum value, confirming the shift to P limitation suggested by the NPP data (Fig S1, Table S9). Soil organic P cleaving does not occur in the P-treated plots of either grassland.

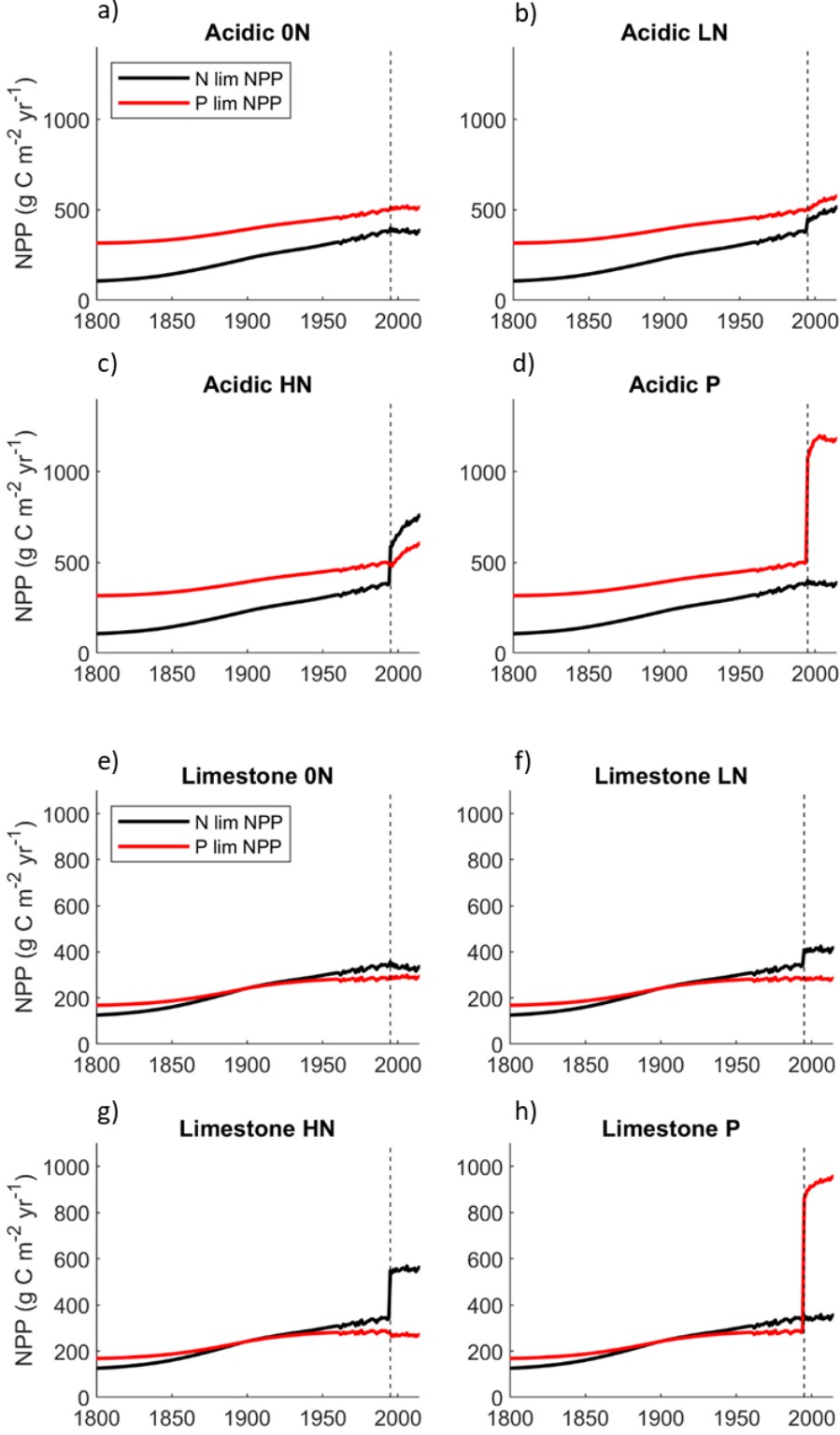

**Figure 3:** Plots showing the nutrient most limiting productivity for all nutrient treatments in both simulated grasslands. The vertical dashed line is the year of first nutrient addition within the model (1995). The value of the lines represents the maximum amount of productivity attainable given the availability of N and P separately. Due to a Liebig's law of the minimum approach to plant growth, it is the lowest of the two lines that dictates the limiting nutrient of the grassland and represents actual modelled productivity. Where lines share a value, it can be considered in a state of N-P co-limitation.

**3.3. Modelled trends and responses to nutrient additions**



The model allows the temporal trends and responses to nutrient additions to be further explored.
Figure 4 provides the temporal responses for the treatments, and Figure 5 a full nutrient budget for
the year 2020. Full data for changes in soil C, N and P and plant biomass C pools since the onset of
large-scale N deposition (1800 within the model) for both grasslands are included in supplementary
Table S14. All data used for determining responses of biomass C and soil organic C, N and P pools to
experimental nutrient additions are in supplementary Tables S15 (acidic) and S16 (limestone).

3.3.1. Acidic grassland
The modelled time series suggest that in the 0N (control) treatment for the acidic grassland,
background levels of atmospheric N deposition between the period 1800-2020 resulted in an almost
four-fold increase in biomass C, a near-twofold increase in SOC and SON and increased the size of
the SOP pool by almost a fifth (Fig 4).
Since initiated in 1995, all C and N pools responded positively to N but not P treatments (Fig 5a, c,
Tables S7, S8). The LN and HN treatments further increased aboveground biomass C by 36.2% and
61.7% (Fig 4a) and increased the size of the total SOC pool by 11.5% and 20.6% respectively (Fig 4c).
Similarly, the total SON pool in the acidic grassland increased by 9.7% in the LN treatment and 36.6%
in the HN (Fig 4e).
Responses of the SOP pool are in contrast to those of the SOC and SON pools, with LN and HN
decreasing SOP by 4.4% and 9.1% respectively, while P addition substantially increased the size of
the SOP pool by 76.7% (Fig 4g).  Nitrogen treatments facilitated access to SOP from both subsoil and
topsoil, increasing plant available P and facilitating its uptake into biomass material (Fig 5e, Table
S8).

3.3.2. Limestone grassland
Model simulations for the limestone grassland also suggest N deposition between 1800 and 2020
considerably increased aboveground biomass C, SOC and SON pools (Fig. 4), but to a lesser extent
than in the acidic grassland. Soil organic C and SON increased by almost half and biomass C more
than doubled. Soil organic P accumulated at a faster rate than in the acidic grassland, increasing by
about a third (Fig 4, Table S14).
Responses of the aboveground biomass C and SOC pools in the limestone grassland differ greatly to
those of the acidic, declining with N addition and increasing with P addition (Fig 4). This response
was ubiquitous to all C pools, with declines in subsoil, topsoil and biomass C (Fig 5b, Table S10).
Biomass C declined by 2.4% and 7.3% with LN and HN addition (Fig 4b) and SOC declined by 0.5%
and 1.4% with the same treatments (Fig 4d). Phosphorus addition increased biomass C and SOC by
22.0% and 6.1% respectively (Fig 4b, d).
Nitrogen treatments increased the size of subsoil, topsoil and available N pools, but led to small
declines in biomass N (Fig 5d, Table S11) The P treatment slightly reduced subsoil and topsoil SON
compared to the control yet increased available N and biomass N, to the extent where biomass N is
greater in the P than HN treatment (Fig 5d, Table S11) Total SON increased by 6.4% and 15.0% with
LN and HN respectively and declined by 0.2% with P treatment (Fig 4f).
The response of the limestone P pools mirrors that of carbon, with declines in subsoil SOP, topsoil
SOP, available P and biomass P with LN and HN addition (Fig 5f, Table S12). The limestone grassland
SOP pool declined by 0.2% with LN and 0.5% with HN addition, with an increase of 20.0% upon
addition of P (Fig 4h). The P treatment substantially increased total ecosystem P in the limestone
grassland, particularly in the topsoil sorbed pool (Fig 5f, Table S12).




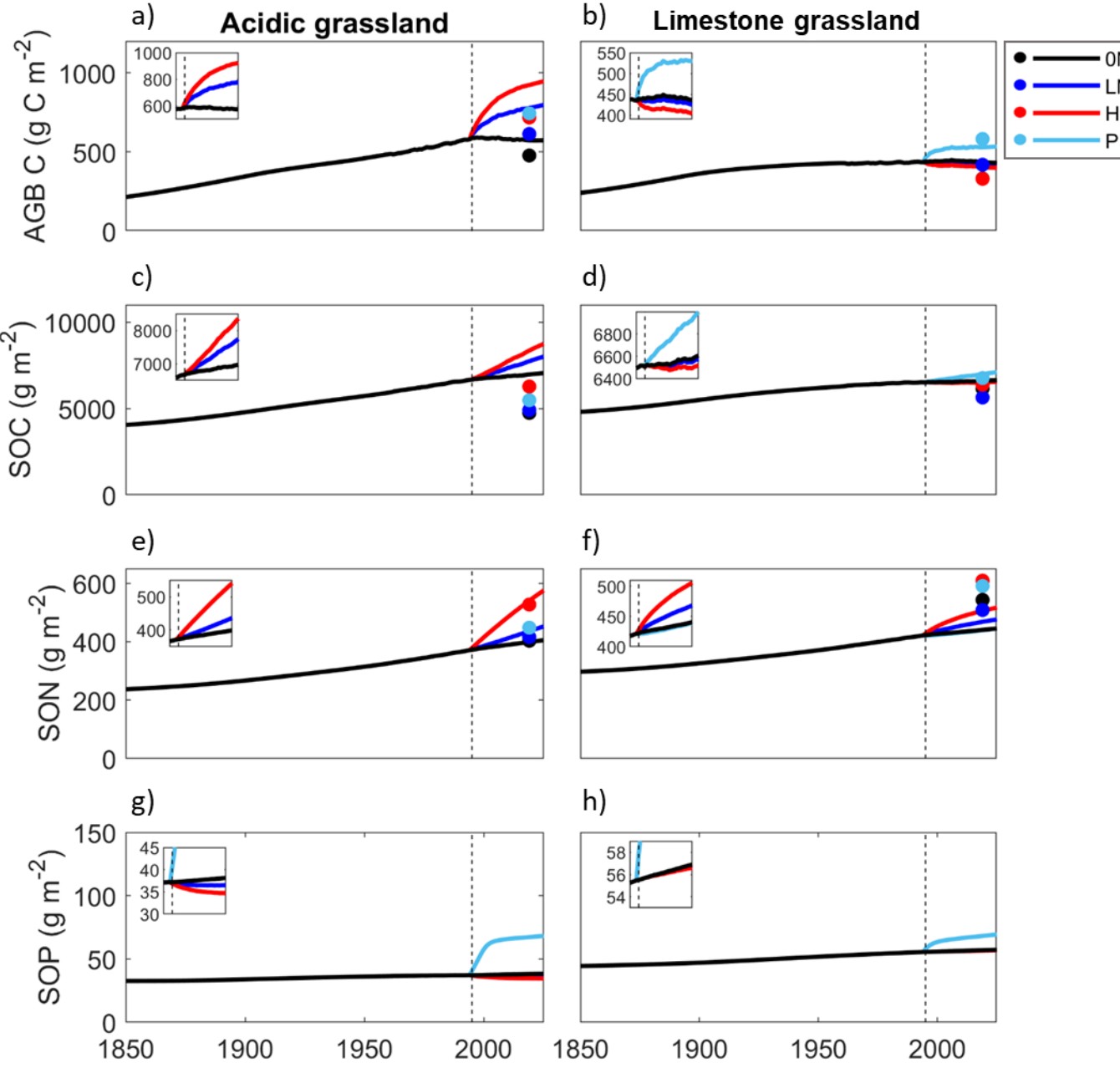

**Figure 4**: Time series plots of aboveground biomass C, soil organic C, N and P for the acidic (panels a, c, e and g respectively) and limestone modelled grasslands (panels b, d, f and h respectively). The vertical dashed line represents the first year of nutrient addition (1995) and marks the beginning of the experimental period. The inset subplots focus on this experimental period (1995-2020) and highlight changes occurring as a result of nutrient additions rather than background N deposition. All nutrient treatments at Wardlow are represented in all panels though not all lines are visible if they do not differ from 0N. Both grasslands share a y axis. Empirical data from figure 2 are plotted on the respective panels, with the exception of panels g and h, where empirical data is incompatible with modelled data (total P versus organic P).

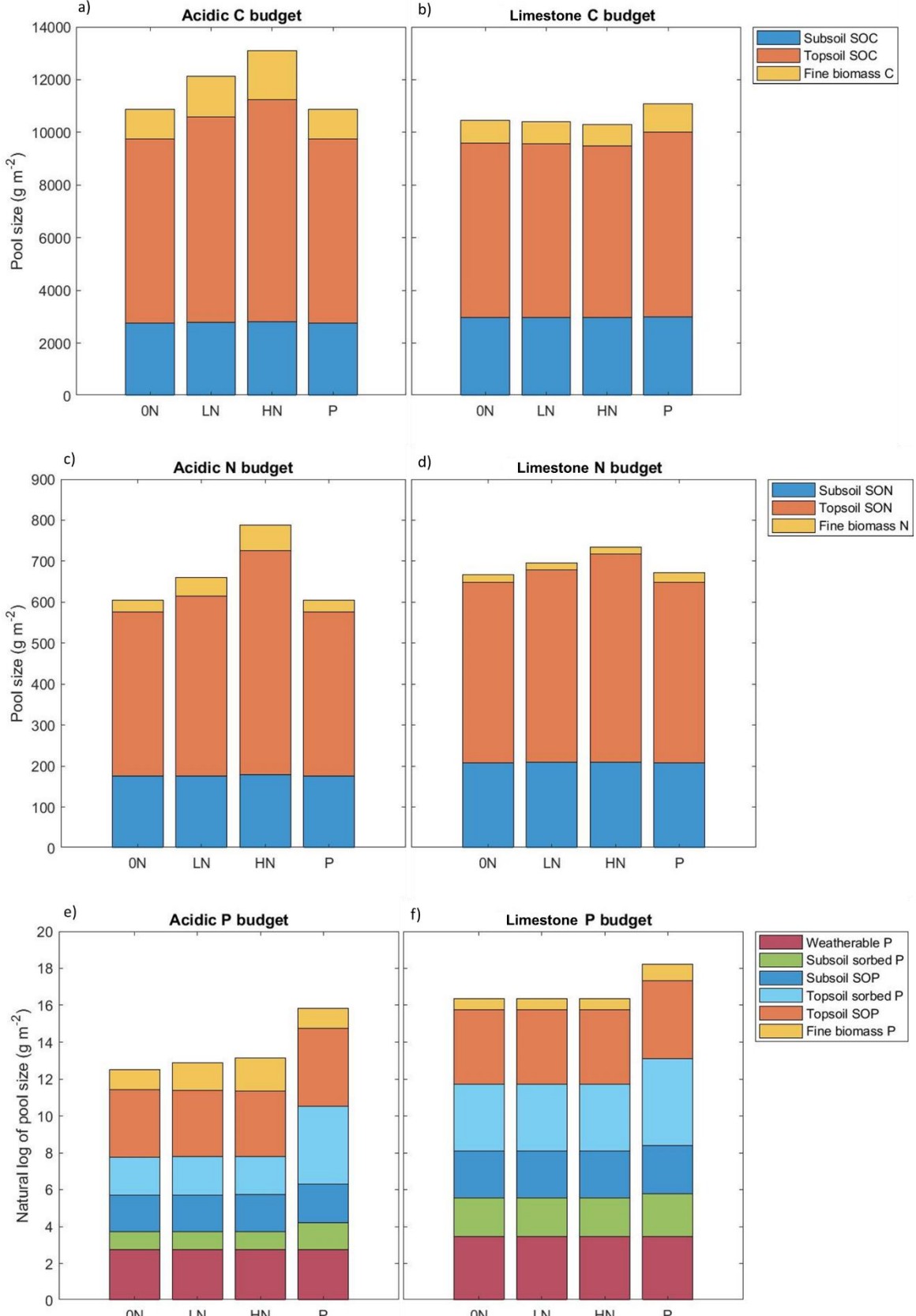

**Figure 5**: Modelled C, N and P budgets for the acidic (panels a, c and e) and limestone (panels b, d, f) grasslands for the year 2020. Modelled sizes of C and N pools are in grams per metre squared, and P pools are presented as $\log_n$ grams per metre squared. Temporary pools such as available N and P and fixed N are not presented here to avoid 'double counting' in other pools and wood litter C, N and P are not presented due to their negligible sizes.

**4. Discussion**
**4.1. Simulating contrasting grasslands by varying plant access to P sources**
This is the first instance in which N14CP, and to the best of our knowledge, any other integrated C-N-
P cycle model, has explicitly modelled P-limited ecosystems and investigated their responses to N
deposition and additional nutrient treatments. By using empirical data from long-term experimental
grasslands to drive and calibrate N14CP, we could test the model's ability to simulate two
contrasting P-limited grasslands, and how organic P access may affect this ability. While the purpose
of this work was not to explicitly reproduce the Wardlow grasslands within N14CP, by comparing
data from Wardlow to the simulated grasslands, we can simultaneously develop our understanding
of the model's representation of under-studied P cycling processes and contextualise what this may
mean for empirical systems such as Wardlow.
The model suggests that the acidic grassland was characterised by high access to organic P, with
comparatively low inorganic P availability, whereas the limestone grassland was the opposite, with
low organic and high inorganic P availability. These simulated differences could reflect the relative
availability of different P sources at Wardlow. As the acidic grassland formed in a hillside depression,
loess has accumulated, thickening the soil profile and distancing the plant community from the
limestone bedrock. The plant rooting zone of the acidic grassland is therefore not in contact with the
bedrock, and roots almost exclusively occur in the presence of organic P sources which can be
cleaved and utilised by plants [Caldwell, 2005; Margalef *et al.* 2017]. Conversely, the limestone
grassland soil rarely exceeds 10 cm depth, and the rooting zone extends to the limestone beneath,
providing plants with greater access to weatherable calcium phosphate [Smits *et al.* 2012].
Such parameter combinations allowed for reasonable congruence between empirical and simulated
data, with an average discrepancy of only 6.6% (SE ± 9.1) and 1.2% (SE ± 4.4) for the acidic and
limestone grasslands respectively across all variables (Table S5). However, model performance
differed greatly between the two grasslands. For instance, the model accurately captured the
magnitude of limestone C, N and P data and their expected P-limited responses to nutrient
treatment, but was less effective at simulating the acidic grassland. N14CP did not simulate an
increase in biomass C or SOC with P addition in the acidic grassland, instead simulating a solely N-
limited grassland. While this may be expected of a model that employs a law-of-the-minimum
approach, N14CP has a number of mechanisms to account for N and P interdependence, meaning
that in principle, it is capable of simulating positive responses to LN, HN and P treatment, as
observed in the empirical data from 2017 (section 2.2.2).
The overestimation of acidic C pools and underestimation of total P suggests that the model is
simulating that too much organic P is being accessed by plants in response to N addition and
transferred into plant biomass pools (Fig 2d). Few parameter sets where simultaneously able to
simulate the magnitude of the empirical TP pool and the positive response of biomass to N addition
in the acidic grassland. This may also be due to limitations in the empirical P data, as P data used for
calibrating P cycling were available for only two nutrient treatments and represented total soil P, not
organic P. While we acknowledge the technical and theoretical issues associated with distinguishing
between organic and inorganic P pools [Lajtha *et al.* 1999; Barrow *et al.* 2020], such distinctions
would help in understanding this discrepancy and likely improve the model's ability to simulate P-
limited systems, particularly when organic P availability may be important.
Additionally, N14CP's representation of organic P cleaving likely underestimates the ability of soil to
rapidly occlude and protect organic P that enters solution. For example, inositol phosphate, a major
constituent of organic P, has been found to be used extensively by plants grown in sand but is hardly
accessed by plants grown in soil [Adams and Pate 1992]. Such organic phosphates become strongly
bound to oxides in the soil, protecting them from attack by phosphatase enzymes [Barrow 2020].
This may be particularly prevalent in the acidic grassland at Wardlow where N deposition has
resulted in acidification and base cation depletion [Horswill *et al.* 2008], potentially enhancing the
formation of iron and aluminium complexes and immobilising P [Kooijman *et al.* 1998].
In addition to physico-chemical processes reducing P availability, in P-limited grassland soils,
microbial processes may be dominant drivers of ecosystem P fluxes [Bünemann et al. 2012]. For
instance, while mineralisation of organic P may increase inorganic P in solution [Schneider *et al.*
2017], this can be rapidly and almost completely immobilised by microbes, particularly when soil P
availability is low [Bünemann et al. 2012]. As the model lacks a mechanism for increasing access to
secondary mineral P forms comparable to organic P-cleaving, and microbial P immobilisation is
incompletely represented for P-limited conditions, it is possible that the uptake of organic P by the
acidic grassland in the model is exaggerated.
The model's inability to simulate a positive response to both N and P addition in the acidic grassland
may be an unintended consequence of the downregulation of N fixation by N deposition included
within N14CP [Davies *et al.* 2016b]. While this representation is appropriate [Gundale *et al.* 2013],
when N deposition exceeds fixation (as at Wardlow), fixation is essentially nullified (as in Tables S7,
S11), meaning deposition becomes the sole source of N to the grassland. This in effect, removes the
dependence of N acquisition on P availability, and could make modelling behaviour akin to N-P co-
limitation [Harpole *et al.* 2011] under high levels of N deposition challenging. This suggests that
current C-N-P cycle models that employ a Liebig's law of the minimum can provide a broad
representation of multiple variables by calibrating access to both organic and inorganic P sources
[Davies *et al.* 2016b], provided the ecosystem in question's limiting nutrient leans towards N or P
limitation. Furthermore, where access to organic P forms is likely to be lower, as in the limestone
grassland, model performance may improve. This could be further explored by allowing N fixation
limits in the model to adapt to P nutrient conditions or by attenuating the suppression of N
deposition on N fixation, to represent acclimatisation of N-fixers to greater N availability [Zheng *et*
*al.* 2018].
Ultimately, differences in modelled accessibility to organic forms of P enabled N14CP to distinguish
between the two empirical grasslands, and simulate the magnitude and pattern of data with
reasonable accuracy, albeit with the previously mentioned caveats.

**4.2. Consequences of differential P access on ecosystem C, N and P**
While the model's estimation of $P_{CleaveMax}$ for the acidic grassland is likely overestimated, the model
experiment has highlighted that differences in organic versus inorganic P availability are a key
determinant of an ecosystem's nutrient limitation, and consequently, how they respond to changes
in anthropogenic N and P availability. For instance, while being exposed to the same background
level of N deposition and the same magnitude of experimental treatment, the modelled acidic
grassland was able to stimulate growth in response to LN and HN treatment whereas the modelled
limestone grassland was negatively affected by it.
Nitrogen addition increases plant demand for P and can shift ecosystems toward a state of P
limitation or increase the severity of limitation where it already exists [Menge and Field, 2007; An *et*
*al.* 2011; Goll *et al.* 2012]. Consistent with this, both simulated grasslands saw SOP decline with LN
and HN treatment, worsening P limitation in the limestone grassland, and depleting the SOP pool in
the acidic. As P cleaved from organic pools is the least bioavailable within the model hierarchy
(methods 2.2.3), this is indicative of increasing P stress in both grasslands. While SOP declined in
both grasslands, the responses of available and biomass P to nutrient treatments differed markedly
between the grasslands. Due to the higher rate of $P_{CleaveMax}$ in the acidic grassland, more P was in
plant-available forms and hence P does not become the limiting factor under N treatments (Table
S8). Conversely, available and biomass P decline under LN and HN addition in the limestone
grassland (Table S12), highlighting how the grassland's $P_{CleaveMax}$ capability is insufficient to meet
increased P demand.
Such high access to organic P sources in the modelled acidic grassland likely led it to respond to
nutrient enrichment in an N-limited manner, increasing productivity in response to N deposition and
LN and HN treatments as the model's limiting nutrient stimulated plant growth. Detrital C inputs
from plant biomass are the primary source of SOC accumulation within N14CP [Davies *et al.* 2016b]
and as such, changes in SOC integrate long term trends in net primary productivity in systems where
external nutrients are supplied. The provision of additional N in the modelled LN and HN treatments
therefore led to large increases in biomass accumulation and consequently, almost linearly increased
SOC (Fig 4c).
Similar increases in N-limited grassland SOC under N addition have been shown, resulting from
significant increases in below-ground carbon input from litter, roots [He *et al.* 2013] and detrital
inputs [Fornara *et al.* 2013], mechanisms similar to those reported by the model. Similarly, Tipping *et*
*al*. [2017] used N14CP to show that N deposition onto N-limited UK ecosystems ubiquitously
increased SOC storage by an average of 1.2 kgCm$^{-2}$ (c. 10%) between 1750 and 2010 [Tipping *et al.*
2017].
Despite its P-limited condition under the HN treatment (Fig 3c), the acidic grassland continued to
accumulate biomass with N addition as the grassland's greater access to topsoil SOP (Table S8)
allowed it to acquire sufficient P to stimulate additional growth but not necessarily to alleviate P
limitation. This is consistent with the acidic grassland at Wardlow, where N treatment stimulated
root surface phosphatases, likely supplying more SOP to plants [Johnson *et al.* 1999]. Our simulated
acidic grassland therefore supports the hypothesis that prolonged N deposition may increase SOP
access to such an extent that P limitation is alleviated and growth can be stimulated [Chen *et al.*
2020]. Organic P release from SOM and its potential immobilisation, is poorly represented in models
and we encourage further study aimed at quantifying these processes [Chen *et al.* 2020; Janes-
Bassett *et al.* 2020; Phoenix *et al.* 2020]. However, such high rates of SOP access only occurred under
experimental LN and HN treatments, and in reality, such rapid degradation of SOP may eventually
degrade the pool to such an extent that P limitation soon returns.
Conversely, biomass C and SOC in the modelled limestone grassland responded positively to P
addition, via similar mechanisms to the N-response in the modelled acidic grassland. However, in
contrast to the acidic grassland, N addition caused declines in limestone biomass and SOC, the
former of which has been observed at the limestone grassland at Wardlow [Carroll *et al.* 2003].
Reductions in limestone biomass C (and consequently SOC) in the model are a combined result of
reductions in bioavailable P (Table S12), occurring via N-driven increases in stoichiometric P demand,
in addition to an inability to access sufficient P from the SOP pool (Table S14). Plants therefore
cannot meet P demand and new biomass is insufficient to replace senesced plant material,
decreasing net biomass C input to the SOC pool. This suggests that in P-limited limestone grasslands
such as at Wardlow, where access to organic P forms may be comparatively limited, N deposition
may worsen pre-existing P limitation and reduce ecosystem C stocks [Goll *et al.* 2012, Li *et al.* 2018].

**4.3. Model limitations**
While N14CP is a fairly simple ecosystem model by design, it is one of few models to integrate the C,
N and P cycles for semi-natural ecosystems and has been extensively tested against empirical NPP
and soil C, N and P data [Davies *et al.* 2016a; Davies *et al.* 2016b; Tipping *et al.* 2017; Tipping *et al.*
2019; Janes-Bassett *et al.* 2020]. Previous work with N14CP has identified the need to enhance its
ability to simulate organic P cycling [Janes-Bassett *et al.* 2020], which we aimed to do in this study by
using long-term experimental data from contrasting P-limited grasslands.
N14CP's simplified representation of plant nutrient pools and plant control over nutrient uptake, is
largely controlled by stoichiometric demand [Davies *et al.* 2016a], and does not incorporate many
plant strategies for P acquisition [Vance *et al.* 2003]. Indeed, by allowing $P_{CleaveMax}$ to vary to account
for empirical data, we attempt to somewhat increase plant control over organic P uptake. We
acknowledge earlier that such an approach likely underestimates the ability of soil surfaces and
microbes to protect newly-cleaved P from plant uptake. As such, where we may expect access to
organic P to be high, such as the acidic grassland at Wardlow, such modelled representation of
plant-mediated P access may lead to unrealistic depletions in soil P and increases in biomass and soil
C, and we would encourage further work aimed at improving model-representation of plant controls
on organic P cycling [Fleischer $et$ $al.$ 2019].
While we feel incorporating a suite of plant strategies for acquiring P would represent over-
parameterisation, we acknowledge that a modelled equivalent to $P_{CleaveMax}$ for accessing inorganic P
forms is lacking, such as carbon-based acid exudation to increase mineral P weathering [Achat $et$ $al.$
2016; Phoenix $et$ $al.$ 2020], which likely contributes toward the poor representation of the acidic
total P pool. Biota-enhanced P weathering and nutrient redistribution by mycorrhizal hyphae are
important for nutrient cycling [Quirk $et$ $al.$ 2012], and fungal community structure and function is
strongly influenced by perturbations in the C and N cycles [Moore $et$ $al.$ 2020]. Such processes are
not included within N14CP as the extent to which weathering can be controlled by such mechanisms
and the manner in which these can be represented in C-N-P cycle models is debated [Davies $et$ $al.$
2016b].
Currently, N14CP assumes C to be in unlimited supply, with its uptake by plants and consequent
input into soil pools controlled by C:N:P stoichiometry, hence C availability has little effect on N and
P dynamics within the model. Increasing atmospheric $CO_2$ may increase nutrient availability, as
plants may reallocate additional carbon resources toward nutrient acquisition [Keane $et$ $al.$ 2020] or
elevated $CO_2$ ($eCO_2$) may increase limitation of other nutrients such as N [Luo $et$ $al.$ 2004]. The
inclusion of $eCO_2$ into N14CP poses a particularly enticing research opportunity, and we aim to use
this study as a foundation for future work to include this process.

## 5. Conclusions

We have shown that by varying two P-acquisition parameters within N14CP, we can account for contrasting responses of two P-limited grasslands and with reasonable accuracy. However, such coarse representation of organic P cycling in the model likely overestimates the ability of plants to use newly-cleaved P and limits our ability to simulate grasslands where N and P interact to control plant productivity, including the potential for N inputs to alleviate P limitation.

Differences in organic P access was a key factor distinguishing the contrasting responses of the modelled grasslands to nutrient manipulation, with high plant access allowing the acidic grassland to acquire sufficient P to match available N from chronic deposition and prevent 'anthropogenic P limitation'. In the acidic grassland, N treatment stimulated plant access of organic P, promoting growth and C sequestration. However, the model suggests that this is an unsustainable strategy, as the SOP pool rapidly degrades, and if N additions are sustained, P limitation may return. Conversely in the limestone grassland, which was less able to access organic P, additional N provision exacerbated pre-existing P limitation by simultaneously increasing plant P demand and reducing P bioavailability. This reduced productivity and consequently C input to soil pools declined, resulting in SOC degradation exceeding its replacement.

We further show that anthropogenic N deposition since the onset of the industrial revolution has had a substantial impact on the C, N and P pools of both the modelled acidic and limestone grasslands, to the extent where almost half of contemporary soil C and N in the model could be from, or caused by, N deposition.

 Our work therefore suggests that with sufficient access to organic P, long-term N addition may alleviate P limitation. Where organic P access is limited, N deposition could shift more ecosystems toward a state of P limitation or strengthen it where it already occurs [Goll *et al.* 2012], reducing productivity to the point where declines in grassland SOC stocks - one of our largest and most labile carbon pools – may occur.

*Data availability:* Data archiving is underway with the NERC's Environmental Information Data
Centre (EIDC) and a DOI will be available once this process is complete. All data to be archived is
present in the supplementary information for review purposes.

*Author contributions:*
CRT: Conceptualistion, data curation, formal analysis, investigation, methodology, project
administration, software, validation, visualisation, writing – original draft preparation, writing –
review and editing
VJB: Conceptualisation, formal analysis, investigation, methodology, supervision, software, writing –
review and editing
GKP: Conceptualisation, methodology, funding acquisition, project administration, resources,
supervision, writing – review and editing
BK: Investigation, methodology, supervision, writing – review and editing
IPH: Funding acquisition, methodology, resources, supervision, writing – review and editing
JD: Conceptualisation, formal analysis, investigation, resources, methodology, supervision, project
administration, software, writing – review and editing

*Competing interests:* The authors declare that they have no competing interests.

*Personal acknowledgements:* We thank Jonathan Leake for his insightful interpretation of our
findings and for constructive feedback on early versions of the work. In addition, we are grateful for
technical assistance from Irene Johnson, Heather Walker and Gemma Newsome, without whom
there would be no carbon and nitrogen data for model input. We are grateful to the Met Office UK
and the Centre for Ecology and Hydrology for use of their meteorlogical and deposition data
respectively.We also wish to extend our thanks to James Fisher for his earlier work on Wardlow
carbon data, which prompted additional investigation into the grassland's carbon stocks. Finally, we
thank the anonymous reviewers for their valuable contributions to improving the manuscript.

*Site access*:  Shaun Taylor at Natural England.

*Funding:* This work was funded by the Natural Environment Research Council award NE/N010132/1
to GKP and NERC award NE/N010086/1 to IPH of the 'Phosphorus Limitation and Carbon dioxide
Enrichment' (PLACE) project. This work was also funded through 'Adapting to the Challenges of a
Changing Environment' (ACCE), a NERC-funded doctoral training partnership to CRT: ACCE DTP
NE/L002450/1.

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
