# Peer review of "Organic phosphorus cycling may control grassland"

_Biogeosciences, 2020_

## Referee Comment (RC1) · Anonymous Referee #1 · 20 Nov 2020

The N14CP model isn't described in much detail, but is reported to have been validated against an extensive range of sites. It appears to be a largely empirical, first-order mathematical approach with climate as the primary plant growth driver, limited by N and/or P availability to meet stoichiometric needs. The rate equations for P uptake and loss are not defined clearly, but the new "cleaving parameter" appears to be a rate co-efficient for a first-order model. The authors finally mention root surface phosphatase enzyme activities on line 539. This should have been done much sooner as a justification for the model formulation and perhaps help inform model formulation. In all, it's

difficult to understand how the model works from the information provided.

The principle focus of simulations seemed to be to derive optimal parameter values that provided best fit to a set of field observations for aboveground biomass (AGB), and soil organic C, N and P pools. The comparisons were difficult to interpret given the scales of simulated vs observed data (Fig. 2). Although the overall relationship between simulated and observed AGB was reasonable, these relationships for soil pools were much weaker and would be better understood if scales were selected to spread the observations. From these data, there was no apparent relationship between observed and simulated values of C and P.

Given the weak validation test results, the lengthy discussions of many simulation patterns and details seem overemphasized. Many of the results are highly speculative and for reasons that aren't clear. For example, the authors discuss many potential interactions between N and P limitations, but how are these explanations based on mechanisms included in the model? On a more general topic, why simulations over such a long time period?

The effects of acidification on P availability regarding iron and aluminum complexes are more complex than referenced on lines 541-545 (see Barrow 2020). More information about how these sites, their mineralogy and pH might influence P availability would help interpret this idea.

In conclusion: this model revision seems to have improved the N14CP model's ability to respond to N and P limitations to plant growth, likely due to adding an organic P source, but the model doesn't capture much of the soil pool dynamics so it could be summarized in a much shorter article.

Barrow, N.J., 2020. Comparing two theories about the nature of soil phosphate. European Journal of Soil Science. https://onlinelibrary.wiley.com/doi/epdf /10.1111/ejss.13027.

---

## Referee Comment (RC2) · Anonymous Referee #2 · 26 Nov 2020

This manuscript investigates the effect of nutrient addition on grassland using a combination of process based modelling and observations from a manipulative experiment. The authors use the simple ecosystem model N14CP to simulate nutrient dynamics at two contrasting grassland sites in the UK and compare these results to data from a long term n and P addition experiment at the same sites. The study shows that P availability difference between the sites leads to differing limitation over time and differing effects of nutrient addition. The question of N and P limitation and co-limitation is very important and very topical, especially in the context of anthropogenic N deposition. Using

process based models in conjunction with manipulative experiments is a very useful tool, not only for validating models but for advancing our understanding of ecosystem processes.

One of the main issues of the manuscript in its current form is the way the observations are actually used to inform the model. As far as I can tell, the data is simply used to calibrate two parameters and then hardly ever mentioned again. The first problem here is the calibration itself: all experimental data is used at this step. This means that implicitly the model can represent observations from all treatments and lowers my confidence in the model's ability to predict responses to nutrient additions. I would suggest performing the calibration with the data from the control plots only, if there is sufficient data.

The second data issue is the lack of model data comparison beyond figure 2. Specifically for figure 4 I wonder if it would be possible to add the observations to the plots rather than just referring to a supplementary table. It might even be useful to just show the experimental period, currently in figure insets, to better show how the model compares to the experimental results. This would increase our confidence in the model and build up the argument towards the predicted long term trends.

I also find that the paper lacks a discussion of the model's short-comings. While I understand the usefulness of simple models in allowing easier process attribution and avoiding over-parameterisation, N14CP is lacking a number of processes compared to state of the art vegetation models and this needs to be acknowledged. Most importantly, model NPP does not appear to include a response to increased CO2. This is particularly important for predictions of long-term nutrient limitation as elevated CO2 has been shown to increase plant nutrient limitation and I do not see how we can have any model predictions that do not take this into account.

Other missing processes are less important, but would still need a paragraph in the discussion, especially the very simple plant pool structure and limited plant control on

nutrient demand and uptake. It is even unclear to me if there is a belowground plant pool that would determine the N and P uptake or indeed if NPP scales with biomass. I do want to stress that I am not suggesting that authors modify their model or that the model is wrong, but that the assumptions in the model structure need to be highlighted and discussed.

―――――――――――――――――――――

---

## Referee Comment (RC3) · Anonymous Referee #3 · 6 Dec 2020

The authors presented a modelling application of the N14CP model to investigate the role of N deposition on the soil C storage of P limited grasslands. The N14CP model seems to be a simple but heavily calibrated model, but it is not adequately described in the paper for the readers to fully understand the long discussion about the pattern of different model outputs.

However, the main deficiency that I find is the model performance against measurements in figure 2. First of all, I don't think the 1-to-1 point plot is the best way to display the results, since each point is representing a different experiment and to me it is more

interesting to see the different model performance of varying scenarios rather than looking at a overall r2 of eight very different scenarios. One example as the authors have already noticed, is that AGB carbon and soil C are noticeably overestimated in acidic grassland but soil N is not, and more surprisingly, total soil P is underestimated. This pattern really indicates that the model is not capturing the SOM stoichiometry, and it actually worries me about the main focus of the paper is on effects of N and P on soil C storage. Secondly, it is unclear to me if all the eight experiments are calibrated or only the two unfertilized ones are calibrated. Also, the initial soil pool sizes are not clear to me either. I find it really difficult to understand how to spin up the model for 10000 years and compare to the present day soil measurement. From figure 3 it seems that the model is still far from equilibrium in both ABG C and soil C, particularly in the acidic grassland. It actually confuses me about the poor soil C correlation between modelled and measured soil C in the acidic soil. Why do you choose to spin up the model for 10000 years, and how does the spin up time affect your results?

A final comment, the discussion need to focus much less on the speculation of model outputs, but include some discussion about the possible caveats of model or study design and uncertainties caused by these limitations.

---

## Author Comment (AC1) · 8 Dec 2020

**Reviewer 1**

Many thanks for your constructive comments on our manuscript. Please find our responses below, with your original comments in regular text and our responses underneath in green:

'The N14CP model isn't described in much detail, but is reported to have been validated against an extensive range of sites. It appears to be a largely empirical, first-order mathematical approach with climate as the primary plant growth driver, limited by N and/or P availability to meet stoichiometric needs. The rate equations for P uptake and loss are not defined clearly, but the new "cleaving parameter" appears to be a rate coefficient for a first-order model.'

We appreciate your comment. Rather than repeat previously published detail we refer to other papers that include more in-depth description as is often the convention with process based modelling. Instead we have attempted to summarise the most pertinent aspects and your interpretation of the N14CP model and the cleaving parameter from the detail provided is accurate. We have perhaps missed some relevant detail in doing so and as you suggest, we could more clearly define the rate equations for P uptake and loss – thank you for this suggestion, we will happily provide this detail from the original model development paper here to aid the reader.

This detail will be added in the '2.2.1. N14CP model summary' section where we discuss plant sources of nutrients

'The authors finally mention root surface phosphatase enzyme activities on line 539. This should have been done much sooner as a justification for the model formulation and perhaps help inform model formulation. In all, it's difficult to understand how the model works from the information provided.'

We agree that explicit and earlier reference to the relevance of the $P_{CleaveMax}$ parameter to root surface phosphatase activity would be useful.

This shall be added and clarified in '2.3.3. Model parameters for the acidic and calcareous grasslands' section.

'The principle focus of simulations seemed to be to derive optimal parameter values that provided best fit to a set of field observations for aboveground biomass (AGB), and soil organic C, N and P pools. The comparisons were difficult to interpret given the scales of simulated vs observed data (Fig. 2).'

While deriving optimal parameter values was an important component of the simulations, the principle focus of the simulations was to explore variation in P acquisition parameters and how these may help account for differing responses to N or P addition. We attempt to find parameter values for these uncertain and under-studied processes by using the observational data from this long-term manipulation experiment.

We appreciate that this focus may be unclear and so will add some clarification in the aims section of the introduction.

With respect to the scales – in these 1:1s we believe retaining the 0 at the lower limit is important as it provides a scale context. We agree however, perhaps the upper limits could be reduced somewhat to narrow the scale slightly and enhance readability.

'Although the overall relationship between simulated and observed AGB was reasonable, these relationships for soil pools were much weaker and would be better understood if scales were selected to spread the observations.'

As explained above, we decided to plot raw (untransformed) observational data against simulated data with x and y axes beginning at 0 to present data as transparently as possible. We previously plotted all data across all variables and both grasslands on the same natural log-scaled axes, which whilst helping the visual spread of data, made the relationship between simulated and observed data appear much more close than in reality, and hence we decided not to use such a format.

We will narrow the upper scales for densely clustered variables (such as SOC and SON) in figure 2 to aid readability as suggested above.

'From these data, there was no apparent relationship between observed and simulated values of C and P'

We acknowledge that in its current state, figure 2 may not be adequately highlighting relationships between observed and simulated values of C and P. We hope that with amendments to the scales and other adjustments that these relationships may be clearer. Where amendments to figure 2 don't help clarify the relationships, we could add more explanation in text.

The relationships that we had hoped to highlight are as follows:

The differences between SOC in the acid plots are similar for simulated and observed, but the magnitude of the estimated pool is higher overall. The model captures the pattern of increasing SOC with N addition fairly well, though for reasons we identify in the manuscript, does not pick up on the increase in SOC with P addition.

For the calcareous plots the magnitude is better captured for SOC and the higher SOC in the P addition plot is picked out by the model. However, unlike with the acidic SOC, the relationship between calcareous SOC and N treatment is less clearly identified, though it is worth noting that there is less difference in the observed SOCs for this site. Taken together, these produce a (misleadingly) small $r^2$ value of 0.01, which would be considerably larger if grasslands where plotted separately.

For P – the magnitudes of the calcareous sites are well captured and there is little difference between the two plots observed. We agree that there is discrepancy between the observed and simulated for the P in the acid site. Why we believe such a discrepancy exists is discussed in the discussion section (lines 508 to 511), and we formally quantify the relationships using $r^2$ and errors in the results section.

'Given the weak validation test results, the lengthy discussions of many simulation patterns and details seem overemphasized. Many of the results are highly speculative and for reasons that aren't clear. For example, the authors discuss many potential interactions between N and P limitations, but how are these explanations based on mechanisms included in the model?'

We hope we have somewhat addressed the overemphasis of this section of the discussion in our previous comment. We do agree that this section is perhaps too lengthy and as you identify, not always attributed to mechanisms included in the model. We will address by either removing parts deemed superlative or speculative or adding explicit references to model processes.

These changes will be made to '4.2. Simulating grassland C, N and P pools by varying plant access to P sources' in the discussion section.

'On a more general topic, why simulations over such a long time period?'

The N14CP model is spun up from the onset of the Holocene to capture the length of time required for soil formation following deglaciation. This is not in an attempt to truly model this long time period but to form an initial condition for modern day simulations that takes in what we know about the site history and forcings. We prefer this method over spinning up a model over an undefined time period until it matches a SOC measurement, as is common practice with other similar models, as it avoids the assumption that soils are presently in steady state (which they are not), and the biasing of results from tuning to that initial stock. If after the spin up period used here, the model can simulate the magnitude of contemporary soil C, N and P pools, it's a good indicator that the processes used by the model and its calibration of initial conditions ($P_{Weath0}$ for example) is suitably reflective of our empirical data.

In addition, N14CP runs on a quarterly time-step and is therefore well-suited to simulating timescales from decades to centuries, which is beneficial considering the timescales of changes in soil pool conditions and nutrient stocks, and responses to long term changes in nutrient availability.

If this comment is more in reference to the timeseries presented in figure 4 then we believe simulating from 1800, and the onset of large scale N deposition across Europe, is the best starting point for investigating the consequences of N deposition on ecosystem C, N and P pools. Additionally, this timescale allows the effect of more recent nutrient treatments to be visually compared to background N deposition effects more clearly.

Thank you for asking this, we expect this to be question other readers may have and so it would be helpful if we were to add some justification into the methodology section. We first mention the timescale in '2.2.2. Net primary productivity' so this would be an appropriate place for more information.

'The effects of acidification on P availability regarding iron and aluminum complexes are more complex than referenced on lines 541-545 (see Barrow 2020). More information about how these sites, their mineralogy and pH might influence P availability would help interpret this idea.'

Thank you for raising this point and for providing a useful reference to help build upon it. We shall add more information about other soil factors that may influence P availability.

'In conclusion: this model revision seems to have improved the N14CP model's ability to respond to N and P limitations to plant growth, likely due to adding an organic P source, but the model doesn't capture much of the soil pool dynamics so it could be summarized in a much shorter article.'

We hope that our previous comments have somewhat justified the detail provided in the article, though we acknowledge that we can reduce areas that may be deemed speculative (parts of the discussion) and that other sections may require additional information (model methods).

---

## Author Comment (AC2) · 8 Dec 2020

**Reviewer 2**

Many thanks for your constructive comments on our manuscript. Please find our responses below, with your original comments in regular text and our responses underneath in green:

'This manuscript investigates the effect of nutrient addition on grassland using a combination of process based modelling and observations from a manipulative experiment. The authors use the simple ecosystem model N14CP to simulate nutrient dynamics at two contrasting grassland sites in the UK and compare these results to data from a long term n and P addition experiment at the same sites. The study shows that P availability difference between the sites leads to differing limitation over time and differing effects of nutrient addition. The question of N and P limitation and co-limitation is very important and very topical, especially in the context of anthropogenic N deposition. Using process based models in conjunction with manipulative experiments is a very useful tool, not only for validating models but for advancing our understanding of ecosystem processes.'

Thank you, we are glad that you agree the research question is important and topical!

'One of the main issues of the manuscript in its current form is the way the observations are actually used to inform the model. As far as I can tell, the data is simply used to calibrate two parameters and then hardly ever mentioned again. The first problem here is the calibration itself: all experimental data is used at this step. This means that implicitly the model can represent observations from all treatments and lowers my confidence in the model's ability to predict responses to nutrient additions. I would suggest performing the calibration with the data from the control plots only, if there is sufficient data.'

Thank you for bringing this to our attention, you raise valid points that we should have clarified in-text. You are correct that the empirical data is used for calibration of $P_{CleaveMax}$ and $P_{Weath0}$ but is seldom mentioned later in the manuscript. More references to the empirical data would certainly be useful for contextualising the results and will be added where relevant (such as figure 4 as you suggest later).

In reference to your point: 'The first problem here is the calibration itself: all experimental data is used at this step', we should highlight that we did exclude empirical biomass carbon data from the cost function to assess the model's ability to simulate empirical data that weren't provided. These outputs are provided in figure 2 panel a. As biomass C is the most responsive of the variables to nutrient additions (both in terms of rapidity and magnitude of response), we deemed this to be the most informative data to set aside for separate model testing.

However, we acknowledge that we are not sufficiently clear and are even contradictory in the methodology section for this where we state:

'The sum of the absolute errors between modelled and observed plant C and soil C, N and P data were scaled (to account for differing numbers of observations) and summed to provide an F value (Equation 1) as an overall measure of error across multiple observation variables.'

Whilst equation 1 does not contain information pertaining to biomass C, we imply in the text above that it is used and we only explicitly state that it isn't in supplementary section S1.1. We shall amend the text accordingly to make it clear that biomass C data was excluded and later used for blind

testing. These changes will be made towards the end of '2.3.3. Model parameters for the acidic and calcareous grasslands'

Regarding using just control data for simulations, we would agree with you that this approach would certainly be sensible and ordinarily deemed most appropriate for a model development study. However, we chose not to do this as the focus of this study is more explorative than it is model development. It was more important to us that the model could capture the potential differences in responses of each nutrient treatment, more so than how well it could simulate the sizes of C, N and P pools in each.

As our main aim was to explore how variation in P access mechanisms helps explain system responses to nutrient perturbations, we thought it better to include data relating to all nutrient treatments in the calibration. If we were to only include control plots, we may have missed the effects of P limitation being exacerbated by N addition captured by the empirical experiment. This is also why we chose to exclude empirical biomass data from all treatments from the initial calibration - to ensure that we could use the calibrated model to simulate responsive variables under all nutrient conditions.

It may be helpful if we provided some justification for this approach in the methodology section '2.3.3. Model parameters for the acidic and calcareous grasslands', so thank you again for raising this.

'The second data issue is the lack of model data comparison beyond figure 2. Specifically for figure 4 I wonder if it would be possible to add the observations to the plots rather than just referring to a supplementary table. It might even be useful to just show the experimental period, currently in figure insets, to better show how the model compares to the experimental results. This would increase our confidence in the model and build up the argument towards the predicted long term trends.'

We have previously explored adding the observations to figure 4 and decided to omit to focus on time series results instead. We could certainly add these back into the figure. We will also adjust this figure to provide greater focus on the experimental period, but still see the value in including time periods before this as it provides context for the magnitude of change and relative drivers of change (i.e. background N deposition Vs experimental additions).

'I also find that the paper lacks a discussion of the model's short-comings. While I understand the usefulness of simple models in allowing easier process attribution and avoiding over-parameterisation, N14CP is lacking a number of processes compared to state of the art vegetation models and this needs to be acknowledged. Most importantly, model NPP does not appear to include a response to increased CO2. This is particularly important for predictions of long-term nutrient limitation as elevated CO2 has been shown to increase plant nutrient limitation and I do not see how we can have any model predictions that do not take this into account.'

Thank you for your comment, and we will expand our discussion of model shortcomings – indeed, every model has short comings. Many models that include the effect of elevated CO2 have major short comings in their inclusion of nutrient effects. Hence, we have focused here on one of the few

models that does integrate carbon, nitrogen and phosphorus cycles for this study that centres on N-P limited grasslands. We agree that elevated CO2 does present an interesting problem – the research team leads such a MiniFACE experiment, to simulate ecosystem responses to concurrent $eCO_2$ and nutrient limitation. However, this study addresses an experiment under past/recent conditions where perturbations in N and P availability far outstrip natural changes in background CO2 concentration over the period, and hence we feel the use of this model is appropriate. This study provides a foundation for future work that could include CO2 effects. We will acknowledge this in the limitations discussion (see below).

'Other missing processes are less important, but would still need a paragraph in the discussion, especially the very simple plant pool structure and limited plant control on nutrient demand and uptake. It is even unclear to me if there is a belowground plant pool that would determine the N and P uptake or indeed if NPP scales with biomass. I do want to stress that I am not suggesting that authors modify their model or that the model is wrong, but that the assumptions in the model structure need to be highlighted and discussed.'

These are important processes to discuss in further detail that we shall amend accordingly, thank you for highlighting them. We shall add a separate sub section toward the end of the discussion to highlight these limitations.

In particular, we shall include detail on the potential effects of $CO_2$ enrichment on N and P availability, how these may be important and why they are currently omitted from N14CP. In addition, we shall discuss the simplicity of the plant pool structures and N14CP's simulation of plant control over nutrient uptake, and add clarification where required.

For example, and in reference to your comments, N14CP does have a belowground plant pool, represented as the root fraction of plant biomass. This was perhaps unclear as we didn't want to include too much repetition from the initial model development paper, which goes into great depth about model processes.

Plant biomass N and P content is somewhat determined by plants via different plant functional type transitions and changes in stoichiometry relative to specific plant end members which differ in their C:N ratios. These are responsive to environmental changes such as nutrient manipulation and / or N deposition. We aimed in this study to in part better represent plant control on P availability by varying $P_{CleaveMax}$ though this was not an explicit aim but may be worth mentioning.

Thanks again for raising these useful points.

---

## Author Comment (AC3) · 7 Jan 2021

**Reviewer 3**

Many thanks for your constructive comments on our manuscript. Please find our responses below, with your original comments in regular text and our responses underneath in green:

'The N14CP model seems to be a simple but heavily calibrated model, but it is not adequately described in the paper for the readers to fully understand the long discussion about the pattern of different model outputs'

Thank you for highlighting this issue, this is also something picked up by reviewer one. To reiterate some of our response there, we attempted here to highlight the overall workings of the model and most relevant processes here rather than repeat the full details of the model which have been published elsewhere, as is common practice with modelling research. We direct the reader to previous papers where the model is described fully, but we appreciate your comments and will seek to expand some of the salient description in this manuscript. We will add more detail to the model methodology section, in particular regarding P uptake and loss and by reducing the discussion of model outputs where these do not explicitly relate to model processes.

More detail regarding P processes will be added in '2.2.1. N14CP model summary' and explicit reference to the relevance of phosphatase enzymes to the $P_{CleaveMax}$ parameter in '2.3.3. Model parameters for the acidic and calcareous grasslands'. Superfluous detail will be cut from '4.2. Simulating grassland C, N and P pools by varying plant access to P sources'.

We would agree that the model is fairly simple by design, though not that it is heavily calibrated. In the application of the model in this study, we calibrate only the $P_{CleaveMax}$ parameter and the initial pool of weatherable P ($P_{Weath0}$). Aside from these two values, all other model parameters are not calibrated to the experimental site.

'However, the main deficiency that I find is the model performance against measurements in figure 2. First of all, I don't think the 1-to-1 point plot is the best way to display the results, since each point is representing a different experiment and to me it is more interesting to see the different model performance of varying scenarios rather than looking at a overall r2 of eight very different scenarios.'

Thank you for your comment. We understand your point of view here, and agree that a one-to-one plot is not the only way the model performance against measurements can be displayed and communicated. We are happy to incorporate the observations into the timeseries plots to help the reader compare the data and model by scenario.

However, we believe the 1-to-1point plot is the most concise way to present data visualising comparisons of both sizes of simulated versus observed pools and to a lesser extent, how they change with experimental nutrient manipulation. The colour coding and markers are intended to help the reader see the different model performance of varying scenarios. The $r^2$ then gives an indication of performance across grasslands and treatments and is an interesting measure as it considers how much the model captures variability across sites/treatments. We agree though that achieving a high $r^2$ is not the ultimate purpose of the study, and that the performance needs to also be interpreted on a 'scenario' basis. We will add a point of clarification on interpretation of the $r^2$ in the text, and ensure we have sufficiently discussed the performance on a site-by-site basis.

'One example as the authors have already noticed, is that AGB carbon and soil C are noticeably overestimated in acidic grassland but soil N is not, and more surprisingly, total soil P is underestimated. This pattern really indicates that the model is not capturing the SOM stoichiometry, and it actually worries me about the main focus of the paper is on effects of N and P on soil C storage'

Thank you for raising this issue, these are important points to discuss.

Firstly, we acknowledge that the overestimation in biomass C and soil C alongside an underestimation in total P may imply that the model has failed to accurately capture all elements of the empirical acidic grassland. There were combinations of $P_{CleaveMax}$ and $P_{Weath0}$ that produced simulated C, N and P pools closer to the empirical pool sizes than the pair of values presented in the manuscript section '3.1. Varying phosphorus source parameters'.

However, we chose to not use this parameter combination as the resulting simulated grassland was behaving in accordance with a solely P-limited grassland rather than the N-P co-limited grassland we know it to be from the empirical data. This was problematic as the empirical data show strong and clear patterns of increasing biomass and SOC with addition of N, which would not have been captured if we used the parameter combination that produced the least discrepancy between the sizes of the observed and simulated pools.

Instead, we used the parameter combination that reduced the discrepancy between the observed and simulated data the most whilst still maintaining behaviour consistent to stronger N than P limitation.

Secondly, as the $P_{CleaveMax}$ parameter is poorly constrained to empirical data, due to comparatively few studies quantifying plant access to organic P, it is possible that the upper limit of $P_{CleaveMax}$ that we set in the calibration is too high. This could explain the pattern you have identified, as plants in the acidic grassland can access more organic P than they perhaps should and use it to stimulate additional growth, leading to reductions in soil P and increases in plant and soil C.

The effect of a potential overestimation of $P_{CleaveMax}$ on SOM stoichiometry may be a limitation of the modelling approach, that needs to be discussed in more detail. We don't believe however that this suggests the model is incapable of capturing SOM stoichiometry, but rather it reflects a relatively poor quantification of organic P cycling in semi natural ecosystems.

We shall discuss these considerations in a dedicated model limitation section in the discussion and will further clarify in text what impact this may have on our understanding of carbon storage for the acidic grassland. Further detail regarding our choice of parameter combinations will be included in the methods section '2.3. Simulating the field manipulation experiment with the model'.

Thank you again for raising this as we feel we have perhaps not explained this sufficiently in the manuscript. Accordingly, we shall expand upon this in the aims section and the methodology section to make it apparent to the reader.

'Secondly, it is unclear to me if all the eight experiments are calibrated or only the two unfertilized ones are calibrated.'

Data from all nutrient treatments within the experiment were included in the initial calibration. This was a concern shared by reviewer two but we feel our approach is justified:

We included SOC, SON and total P data from the 0N, LN, HN and P treatments into the cost function to determine optimal P cycling parameters. However, we excluded all biomass data across all four nutrient treatments to be used for separate blind model testing. We decided to use the variable that responded most rapidly and variably to nutrient additions to test the calibrated model, as this would have provided the most robust possible test with the available data.

Ordinarily we would agree that using only unfertilised data for the calibration would be most appropriate for a model development study. However, we should emphasise that this study was more exploratory than developmental and as such it's necessary to use data from all the various treatments to explore these uncertain variables.

'Also, the initial soil pool sizes are not clear to me either.'

Apologies but we are unsure what is meant by 'initial'. Can you please expand on this?

The only soil pool initialised in this study is the $P_{Weath0}$ condition, which represents the initial pool of weatherable P upon soil formation. Upon mineral weathering, this enters more available soil P pools and can become available to plants and so is an important determinant not only of ecosystem nutrient limitation, but also for determining contemporary C, N and P pools.

The calibrated $P_{Weath0}$ pools are provided for each grassland in the results '3.1. Varying phosphorus source parameters' lines 363 – 364. There is no initial pool of C and N at the beginning of soil formation.

'I find it really difficult to understand how to spin up the model for 10000 years and compare to the present day soil measurement. From figure 3 it seems that the model is still far from equilibrium in both ABG C and soil C, particularly in the acidic grassland.'

This is the benefit of spinning up over time rather than a 'spin up to equilibrium' approach. Real ecosystems are rarely in equilibrium due to constantly changing multiple conditions and so our approach avoids this assumption.

To allow this spin-up, as described in the paper, we use a variety of input data. Inputs nearer the present are more accurately defined based on site-scale measurements, and assumptions are made regarding past conditions:

- Climate: Site based temperature and precipitation data is used for the past 60 years, and prior to this, mean annual temperature was temporally varied using an anomaly based on Davis *et al.* (2003) and mean annual precipitation was maintained as constant
- N deposition: Data regarding Wardlow-specific N deposition from 2004 to 2014 was incorporated and scaled using the historical anomaly formulated by Schopp *et al.* (2003), in order to simulate site-scale background deposition.
- Land use history: A land cover history is defined that sets the plant cover type on an annual basis in the model. This was set using pollen stratigraphy data for the sites spanning the majority of the spin up phase.

We would be happy to add further clarifying details on this and the rationale for this spin-up approach (which is not a new approach) in the text.

'It actually confuses me about the poor soil C correlation between modelled and measured soil C in the acidic soil.'

Thank you for raising this, the correlation between observed and simulated SOC certainly does appear poor at 0.01. This is likely because we have grouped the two grasslands together in the regression analysis, which may otherwise yield more reasonable $r^2$ values if calculated separately for each grassland (with the caveat of having half the data points). We can certainly explore calculating these regressions separately to see if it helps clarify the relationship between modelled and measured SOC.

'Why do you choose to spin up the model for 10000 years, and how does the spin up time affect your results?'

This relates to the previous comment on spin-up above, and was also asked by reviewer one, emphasising that we should certainly add more justification to the methodology section here (section '2.2.2. Net primary productivity'). We refer to our response below:

'The N14CP model is spun up from the onset of the Holocene to capture the length of time required for soil formation following deglaciation. This is not in an attempt to truly model this long time period but to form an initial condition for modern day simulations that takes in what we know about the site history and forcings.

We prefer this method over spinning up a model over an undefined time period until it matches a SOC measurement, as is common practice with other similar models, as it avoids the assumption that soils are presently in steady state (which they are not), and the biasing of results from tuning to that initial stock. If after the spin up period used here, the model can simulate the magnitude of contemporary soil C, N and P pools, it's a good indicator that the processes used by the model and its calibration of initial conditions ($P_{Weath0}$ for example) is suitably reflective of our empirical data.

In addition, N14CP runs on a quarterly time-step and is therefore well-suited to simulating timescales from decades to centuries, which is beneficial considering the timescales of changes in soil pool conditions and nutrient stocks, and responses to long term changes in nutrient availability.'

In reference to us choosing this time period, this is to capture the length of time required for soil formation following deglaciation in north west Europe around this time. We believe this to be the most appropriate time period to use, especially considering we simulate contemporary pools largely by varying the amount of weatherable phosphorus available at the beginning of this spin up phase.

'A final comment, the discussion need to focus much less on the speculation of model outputs, but include some discussion about the possible caveats of model or study design and uncertainties caused by these limitations.'

Thank you for raising this and for the rest of your comments. We shall be adding a designated section into the discussion to explain some of the model limitations and caveats identified by yourself and the other two reviewers.

Specifically, we shall include detail on:

- The simplicity of the plant pool structures and N14CP's simulation of plant control over nutrient uptake, and add clarification where required, including regarding the $P_{CleaveMax}$ parameter earlier in the methodology section and its potential overestimation.
- The potential effects of CO2 enrichment on N and P availability, how these may be important and why they are currently omitted from N14CP.
- Limitations regarding the quarterly time step used by the model (that allow us to spin up from 10,000 years ago) will be discussed
- The key limitation regarding N-P co-limitation in a model using a Leibig's law of the minimum approach, which we believe may be leading to some of the previous patterns you identify.
- Additional considerations of caveats / model simplifications such as the subsurface transferal of nutrients via fungal networks and the flexibility of plant stoichiometry

Other clarifications in-text will include justification for a calibration using all experimental treatments and some clarification that the simulated grasslands are better considered as models of N and P limited semi-natural ecosystems based on empirical data, rather than perfectly modelled representations of the empirical grasslands.

**Reference**

Davies, J. A. C., E. Tipping, E. C. Rowe, J. F. Boyle, E. G. Pannatier, and V. Martinsen (2016), Long term P weathering and recent N deposition control contemporary plant-soil C, N, and P, Global Biogeochemical Cycles, 30(2), 231-249. https://doi.org/10.1002/2015GB005167

Davis, B. A. S., S. Brewer, A. C. Stevenson, and J. Guiot (2003), The temperature of Europe during the Holocene reconstructed from pollen data, Quat. Sci. Rev., 22(15–17), 1701–1716

Schöpp, W., M. Posch, S. Mylona, and M. Johansson (2003), Long-term development of acid deposition (1880? 2030) in sensitive freshwater regions in Europe, Hydrol. Earth Syst. Sci. Discuss., 7(4), 436–446

---

## Author Response (AR1)

**Revisions documentation**

**General comments**

Firstly, we thank all three reviewers for taking the time to review our manuscript, and for offering constructive criticism and suggestions that have allowed us to improve it, as detailed in the reviewer-specific responses below.

There were a number of concerns/suggestions that were shared by multiple reviewers, leading us to pay particular attention to addressing these. These include:

1) More detail on the N14CP model, how it is used, and a discussion of model limitations;
2) Removal of unnecessary detail from the discussion;
3) A number of figure improvements; and
4) Clarifying the narrative throughout the manuscript to ensure that our initial intentions and aims of the work are more clearly communicated.

With respect to this fourth point, we hope that it is now clear that the aim of this work is to explore if we can account for empirical biomass C and soil C, N and P stocks under varying nutrient conditions in P-limited grasslands, using our conceptualisation of P access mechanisms within the model. This allows us to test our understanding of these ecosystems and how we have embedded that understanding in models, providing insights that can help guide future empirical and modelling research. The clarified narrative of the revised manuscript is now better reflected by the new title, which no longer centres on the suggested implications for SOC storage.

We believe this is more appropriate because, while there are clearly shortcomings in the model's ability to simulate these grasslands, we hope to have clarified that these disagreements can help inform future model development, and help identify N14CP's 'current state' abilities, with implications for the simulation of C:N:P dynamics in other ecosystems and Earth system models.

We emphasise that the simulated acidic and limestone grasslands reflect the model's 'best guess' representation of the C, N and P dynamics of the site but that it likely misses some of the empirical nuances. Nevertheless, with acknowledgement of such caveats, these simulated grasslands can be used to explore the potential effects of long term N deposition and nutrient manipulation (as experienced by the Wardlow grasslands), and the potential role(s) P-access mechanisms may play in determining their responses.

Much of the detail we provided in our initial responses to reviewer comments have been included in the revised manuscript text, hence we have not repeated the content of these responses here. Below, we first provide specific changes directly relating to reviewer comments, followed by a description of each revision that has been made.

**Please note:** the referenced line numbers relate to the marked-up version of the manuscript, where tracked changes are visible, and will be incorrect if these are hidden.

Reviewer comments are italicised and quoted, our responses are underneath.

**Reviewer 1**

*'The N14CP model isn't described in much detail, but is reported to have been validated against an extensive range of sites. It appears to be a largely empirical, first-order mathematical approach with climate as the primary plant growth driver, limited by N and/or P availability to meet stoichiometric needs. The rate equations for P uptake and loss are not defined clearly, but the new "cleaving parameter" appears to be a rate coefficient for a first-order model.'*

This broad interpretation of how the model functions is accurate so we are pleased that this was sufficiently communicated within the text. We are happy to provide more detail to aid the reader, especially with respect to P dynamics which are of particular relevance to this work. However, we wish to avoid repeating the previously published full model description in the interest of manuscript length.

We have therefore provided additional detail about the model in the manuscript, as summarised below:

**2.2. Summary of model processes**

2.2.1. N14CP model summary

- Lines 245 - 253
    - We clarify that N14CP has not been explicitly tested for ecosystems exhibiting P limitation, particularly for experimentally-manipulated ecosystems.
    - We also highlight that while weatherable P ($P_{Weath0}$) has been explored within the model, organic P access, and its effects on ecosystems, has not been explored.
- Lines 250 - 253
    - We briefly state what amendments have been made to N14CP to achieve our aims
- References to the initial model development study are still provided in the body of the text for readers wanting additional detail, but we now believe sufficient information is included in the manuscript to fully understand model outputs and discussion.

2.2.2. Net primary productivity and nutrient limitations

- Lines 259 – 280
    - More detail describing how nutrient limitation is determined in the model and how this relates to N and P availability and plant functional type stoichiometry.
- Lines 286 - 293
    - We provide more detail about the relationship between N fixation, PFT and P availability.

2.2.3. Plant and soil N and P cycling

- Lines 297 - 300
    - A summary paragraph describing the updated figure 1 and including information on what detail is excluded from the initial model development paper, and what changes have been made for the purpose of this study (in the figure legend).

- Lines 310 - 313
  - Additional detail about how $P_{weath0}$ contributes toward the plant available P pool, and making it more relevant to plant nutrient limitation.
- Lines 317 - 333
  - Here we have added more detail about P cycling processes including:
    - How the size of the available P pool is calculated, and the hierarchy which the model uses to allow access to these P forms, with cleaved P being the least accessible form.
    - How plants access SOP with information about the $P_{Cleave}$ parameter, including the limits imposed on it by the model.
    - A rate equation for P removal from the SOP pool and how it is limited by SOM C:P stoichiometry (the new equation 1).
    - Discussion about how $P_{CleaveMax}$ differs from the initial $P_{Cleave}$ function within N14CP and how we use it in this work.
- Lines 368 - 378
  - We clarify how C, N and P is lost from the modelled ecosystem.

Figure 1 has been updated to provide a more detailed overview of C, N and P pools and fluxes at the whole ecosystem scale, encompassing atmospheric, plant, topsoil and subsoil / leachate pools.

- We have updated the legend to better explain the processes in the figure and to directly relate to the methodology section 2.2 where we summarise key model processes.
- We further highlight on the figure aspects of N14CP that are of particular relevance to this work, including the provision of experimental N and P and the $P_{CleaveMax}$ parameter.

*'The authors finally mention root surface phosphatase enzyme activities on line 539. This should have been done much sooner as a justification for the model formulation and perhaps help inform model formulation. In all, it's difficult to understand how the model works from the information provided.'*

We have refocussed the narrative of the manuscript to focus more on phosphatase activity and how that relates to our $P_{CleaveMax}$ parameter and organic P cycling within the model.

Phosphatase activity is now included in the introduction on line 115, and it is discussed more in the methods (lines 212 – 213) and discussion sections. It's relevance to model processes has been expanded upon in section 2.3.3. lines 322 - 333, and please see the changes to the model description as detailed in response to previous comment.

*'The principle focus of simulations seemed to be to derive optimal parameter values that provided best fit to a set of field observations for aboveground biomass (AGB), and soil organic C, N and P pools. The comparisons were difficult to interpret given the scales of simulated vs observed data (Fig. 2).'*

We have clarified the aims (lines 167 – 174) and hypotheses (lines 175 – 181), and adjusted the narrative to make it clear that deriving optimal parameter values was an important part of the simulations but was not the principal aim of the work. We provide additional text to clarify our aims at the beginning of relevant sections e.g. lines 250 - 253, 425 – 437 (both methods) and lines 571 – 579 as an introduction to the results section, as well as in the abstract.

The scale on figure 2 has been amended (x and y axes have been reduced) to spread the data points for SOC and SON to aid interpretation of the data.

*'Although the overall relationship between simulated and observed AGB was reasonable, these relationships for soil pools were much weaker and would be better understood if scales were selected to spread the observations.'*

We have addressed this by contracting the axes as much as possible while keeping the lower boundaries at 0 to provide context of scale. This was done for SOC and SON, which were the two most densely clustered variables (Fig 2b, 2c).

*'From these data, there was no apparent relationship between observed and simulated values of C and P'*

We have added detail throughout the manuscript that should aid in interpretation of the relationship between observed and simulated data:

- Methods section 2.3.3. Lines 551 – 565
  - We provide context for what the cost function aims to do and how it helps us interpret the discrepancies between simulated and observed magnitudes of C, N and P stocks.
  - We clarify how we chose the parameter combinations using the F value and what the caveat of using this approach is.
  - In the last paragraph of the methods section lines 559 – 565, we acknowledge that such an approach may represent a compromise for simulating some aspects of empirical data.
- Results section 3.1. has been revised to address reviewer concerns about the apparent lack of relationship between simulated and observed C and P data, to clarify for the reader how we interpret the $r^2$ values:
  - Lines 615 – 636
    - Here we explicitly reference the relationship between simulated and observed data using measures of both magnitude (1 to 1 line) and pattern ($r^2$) of data.
    - We point out where the $r^2$ value may be of little use, such as with low numbers of data points in fig 2d.
    - Where the $r^2$ value appears particularly low for SOC, we describe the model's performance at capturing the pattern in responses of empirical data, which are not picked up by using a regression statistic.

- - The $r^2$ value would likely be higher if we split the grasslands, as the empirical data have opposite directionality, meaning the $r^2$ is likely to be low – however, we felt it inappropriate to run separate regressions on grasslands as this would have resulted in only 4 data points each and instead we chose to describe the relationship in text.
    - The figure legend of figure 2 has been corrected so the $r^2$ value is more appropriately described.
- We address and discuss the poor relationship between simulated and observed acidic P data in the discussion lines 779 - 798, and justify why we believe this to have occurred, what this may mean and suggest ways to improve this aspect of model performance.

- Crucially, the manuscript has been amended to highlight that this apparent lack of relationship is not a failing of the model *per se* as our aim was not to perfectly simulate the Wardlow field site grasslands, but instead gives us insight into how modelled organic P cycling functions, and how it differs from our empirical understanding.

This is now explicitly stated in the methods section 2.3, lines 431 - 437 and is more widely incorporated into the narrative of the paper.

*'Given the weak validation test results, the lengthy discussions of many simulation patterns and details seem overemphasized. Many of the results are highly speculative and for reasons that aren't clear. For example, the authors discuss many potential interactions between N and P limitations, but how are these explanations based on mechanisms included in the model?'*

We agreed with this statement and have addressed it by first providing more detail in the methods section so readers better understand the N-P interactions within N14CP, and by removing any speculative results in the discussion section. All results are now discussed with explicit reference to model processes described in the methods section. This has reduced the length of the discussion and made it much less speculative.

Changes to the methods were highlighted above and the discussion section has been rewritten with the following key changes:

- Repetitive reference to SOP access has been addressed by combining paragraphs that share a similar theme
- As a result, the structure of the discussion has changed to better represent this focus, now consisting of only 2 sub sections (prior to the new limitations section) discussing a) parameter selection and consequent model performance and b) consequences of such combinations on modelled ecosystem C, N and P cycling
- The previous section '4.3. The limiting nutrient through time' has been removed as it was too speculative, though relevant discussion of modelled changes to nutrient limitation have been incorporated elsewhere
- Discussion regarding SON dynamics has been removed as it was of little relevance to previous sections in the manuscript
- Results regarding the model's inability to simulate a positive response to P addition in the acidic grassland have been related to N-P interactions within the model on lines 799 – 813

- Excessive discussion of the modelled declines in SOC stocks has been removed as the cited literature showed contrasting results, hence extrapolating our findings to any one empirical mechanism in particular would have been speculative

*'On a more general topic, why simulations over such a long time period?'*

We choose to spin up the model by simulating since the end of the last glaciation to avoid the issues introduced by assuming the system is in steady state at the start of the period of interest. We include this justification in the methods 2.3.2, lines 449 – 464.

*'The effects of acidification on P availability regarding iron and aluminium complexes are more complex than referenced on lines 541-545 (see Barrow 2020). More information about how these sites, their mineralogy and pH might influence P availability would help interpret this idea.'*

We expand upon this point on lines 789 - 798 and use the reference kindly provided to do so. We highlight how organic phosphates can be bound to oxides in the soil and that this protects them from attack by phosphatases.

*'In conclusion: this model revision seems to have improved the N14CP model's ability to respond to N and P limitations to plant growth, likely due to adding an organic P source, but the model doesn't capture much of the soil pool dynamics so it could be summarized in a much shorter article.'*

We hope the new manuscript better reflects our intentions of modelling the site and our exploration of organic P access. We have expanded on model detail required to interpret our results, and cut content where it does not relate to described model processes.

**Reviewer 2**

*'One of the main issues of the manuscript in its current form is the way the observations are actually used to inform the model. As far as I can tell, the data is simply used to calibrate two parameters and then hardly ever mentioned again. The first problem here is the calibration itself: all experimental data is used at this step. This means that implicitly the model can represent observations from all treatments and lowers my confidence in the model's ability to predict responses to nutrient additions. I would suggest performing the calibration with the data from the control plots only, if there is sufficient data.'*

Thank you for highlighting this important issue to address within the manuscript.

First, we corrected the text that suggested we included all experimental data within the cost function on lines 537 and 541. Previously, we incorrectly stated that plant C data was included in the cost function, which as the reviewer identifies, would have been problematic as no data would have been left for blind testing.

We discuss which data were excluded from the cost function and why in a paragraph spanning lines 551 – 558, where we highlight that we kept biomass C separate as it would provide the most robust test of the model with the available data. We explain that we included data from all experimental treatments (but not data from all variables), to explore if calibrating these two P process parameters could help account for observations across the treatments. Hence, using only control plot data would not allow us to do this.

*'The second data issue is the lack of model data comparison beyond figure 2. Specifically for figure 4 I wonder if it would be possible to add the observations to the plots rather than just referring to a supplementary table. It might even be useful to just show the experimental period, currently in figure insets, to better show how the model compares to the experimental results. This would increase our confidence in the model and build up the argument towards the predicted long term trends.'*

Thank you for this suggestion. We added the empirical data from figure 2 into figure 4, to contextualise the magnitude of the time series with respect to empirical data, and agree it has added value to the revised figure. The figure legends have also been updated to match this. To address the latter comment regarding the experimental period, we have shifted the x axis to focus more on present day simulations, though we still include a period prior to the experiment to better contextualise the changes in response to more recent nutrient additions. We started the x axis 50 years later and extended it a little beyond 2020 so the empirical data was more visible. We also edited the inset subplots to make the experimental period clearer, removing the inset x axis labels, which we felt cluttered the subplot unnecessarily.

*'I also find that the paper lacks a discussion of the model's short-comings. While I understand the usefulness of simple models in allowing easier process attribution and avoiding over-parameterisation, N14CP is lacking a number of processes compared to state of the art vegetation models and this needs to be acknowledged. Most importantly, model NPP does not appear to include a response to increased CO2. This is particularly important for predictions of long-term nutrient limitation as elevated CO2 has been shown to increase plant nutrient limitation and I do not*

*see how we can have any model predictions that do not take this into account.'*

We have added a dedicated section in the discussion to specifically address these comments (lines 1052 – 1087). In this section, we summarise how N14CP has been used previously and in this work, and discuss the limitations of our approach of varying P access conditions to account for empirical data.

With respect to $CO_2$, we agree that accounting for increased carbon availability within biogeochemical models is likely very important for determining future nutrient limitation and N and P cycling, particularly as $CO_2$ concentration continues to increase. Indeed, this is something we hope to explore with N14CP in future work. It would have been useful to know which models the reviewer is specifically referring to as state-of-the-art, as models tend to have varying purposes and foci. For example, whilst DGVMs might contain more plant and carbon process detail, they often have less soil process detail and fewer nutrient interactions. On lines 1081 – 1087, we address the omission of an elevated $CO_2$ effect within N14CP including how carbon availability is currently incorporated in the model and what effects elevating $CO_2$ may have on N and P availability.

*'Other missing processes are less important, but would still need a paragraph in the discussion, especially the very simple plant pool structure and limited plant control on nutrient demand and uptake. It is even unclear to me if there is a belowground plant pool that would determine the N and P uptake or indeed if NPP scales with biomass. I do want to stress that I am not suggesting that authors modify their model or that the model is wrong, but that the assumptions in the model structure need to be highlighted and discussed.'*

As suggested, we have also included discussion of these model limitations in the relevant section. We explicitly address the simplified plant pool structure and plant control over nutrient uptake in lines 1061 - 1070 and how these may be better represented in the future. In addition, on lines 1071 - 1080, we also discuss other potential biological controls on nutrient cycling that are currently lacking in N14CP such as biota-enhanced P weathering and nutrient redistribution and plant control over uptake of other inorganic P forms. On lines 260 – 261 of the methods section, we have highlighted that belowground plant biomass for the fine and coarse pools for each PFT is represented by a root fraction.

**Reviewer 3**

*'The N14CP model seems to be a simple but heavily calibrated model, but it is not adequately described in the paper for the readers to fully understand the long discussion about the pattern of different model outputs'*

The model description was also highlighted by reviewer 1, and we have added more detail to the methodology sections to more fully describe the model, updated the schematic in figure 1 to make it clearer and more detailed, and have removed detail from the discussion where results do not directly relate to described model processes – please see comments above in response to reviewer 1.

With respect to the comment that N14CP is heavily calibrated – we disagree that this is the case. A number of key parameters as identified by sensitivity analyses have been calibrated against data and extensively blind tested in previous publications, which are referenced (Davies et al 2016a,b; Tipping et al 2017; Tipping et al 2019; Janes-Bassett et al 2020). This paper calibrates only 2 parameters to explore the importance of P access mechanisms in determining current plant-soil nutrient pools for grasslands that are of a P-limited nature, as the model has predominantly been used in N-limited settings.

*'However, the main deficiency that I find is the model performance against measurements in figure 2. First of all, I don't think the 1-to-1 point plot is the best way to display the results, since each point is representing a different experiment and to me it is more interesting to see the different model performance of varying scenarios rather than looking at a overall r2 of eight very different scenarios.'*

We justify our use of the 1 to 1 plot in our author response to this comment, but have made some changes to this and other figures and text to help clarify. First, we amended figure 2 by reducing the axes length to spread the data points so patterns are easier to observe and by correcting our description of what the $r^2$ value means in the legend.

Second, and as per reviewer 2's recommendation, we added empirical data into the time series plots in figure 4 which we now believe better represents modelled (and empirical) grassland responses to different experimental treatments. Figure 4 now provides a much clearer representation of grassland responses on a single experimental scenario basis.

Finally, in our discussion of results relating to figure 2 (results section 3.1. lines 604 – 636), we much more clearly interpret what the different $r^2$ values mean and how the model simulates the empirical responses to experimental treatments so interpretation is less dependent on $r^2$ values alone.

*'One example as the authors have already noticed, is that AGB carbon and soil C are noticeably overestimated in acidic grassland but soil N is not, and more surprisingly, total soil P is underestimated. This pattern really indicates that the model is not capturing the SOM stoichiometry, and it actually worries me about the main focus of the paper is on effects of N and P on soil C storage'*

We believe this to be one of the most pertinent comments to address as we did not want our interpretation of results to give the impression that the model cannot accurately capture SOM stoichiometry.

As a result, we have edited the manuscript text to make it clear that this work was not an attempt to perfectly replicate data from the empirical experiment. We use data from the Wardlow experiment to explore the role of differential access to organic P in explaining observed responses - to better inform our understanding of P cycling within the model, and to link this understanding to potential empirical processes at the site.

Where aspects of the experimental data are not well described by the model, this brings to light issues with our current understanding and conceptualisation of ecosystem processes, and this can help inform future empirical work and model development. We explicitly highlight and discuss the overestimation in acidic C and underestimation of P on multiple occasions within the revised version, and explain why we think this occurs and what it means for our understanding. Discussion regarding this aspect of model performance is detailed on lines 779 – 788 and we relate this to empirical processes in the following paragraph lines 789 – 798.

We also note this as a caveat in the model limitations section lines 1061 – 1070 and in the conclusion lines 1093 – 1096 to be as transparent as possible to aid the reader's interpretation of our results. Finally, we have made many changes to the text to remove some of the focus from the impacts of N and P addition on C stocks, as modelling these effects on their own were not the primary focus of the study. We believe the content of the revised manuscript is now better reflected by the new title.

*'Secondly, it is unclear to me if all the eight experiments are calibrated or only the two unfertilized ones are calibrated.'*

This was a concern shared by reviewer 2, so we have made it clearer in text that the calibration included data from all nutrient treatments (and why) but did not include data from all variables. We kept plant biomass C from all nutrient treatments separate for model testing of the selected parameter values.

Specific changes can be found in our response to reviewer 2's similar comments but are summarised below:

- We corrected the lines where we say we included biomass data in the cost function as this was in error – lines 537 and 541.
- We more clearly discuss which data were excluded from the cost function and justify why – 551 – 558.
- We explain why we used the approach of including data from all experimental treatments as opposed to just control – lines 554 - 558

Finally, we acknowledge the limitations of using such a cost function approach and justify why it is still appropriate – lines 559 – 565

*'Also, the initial soil pool sizes are not clear to me either.'*

We are unsure which pools this comment was referring to but we hope that the additional detail we have provided in the methods section addresses this comment. Specifically, we explain in more detail how the initial pool of weatherable P contributes to available P in lines 310 – 313. If this comment is more in reference to the spin up phase of the model simulations, we provide more detail on this on lines 449 – 464.

*'I find it really difficult to understand how to spin up the model for 10000 years and compare to the present day soil measurement. From figure 3 it seems that the model is still far from equilibrium in both ABG C and soil C, particularly in the acidic grassland.'*

This was a concern shared by reviewer 1 so we have added more detail regarding the spin up phase, what the purpose of this phase is (and is not), how we use input driver data to inform us about the site's past history and forcings and how this allows us to simulate present-day measurements. We also justify why this method is appropriate for sites such as Wardlow where anthropogenic nutrient input has been substantial and persistent and hence the ecosystem is unlikely to be in equilibrium.

The reasons for such a lengthy spin up phase are now detailed in methods section 2.3.2. lines 449 – 464, including a justification for this approach. The importance of PFT to nutrient cycling is now referenced earlier on lines 270 – 278 and how we can use PFT input data spanning the entire spin up phase to inform the model is also detailed within the text lines 461 - 464.

*'It actually confuses me about the poor soil C correlation between modelled and measured soil C in the acidic soil.'*

We acknowledge that the use of an $r^2$ statistic on figure 2 may cause some confusion regarding the relationships between observed and simulated SOC. As we did not feel it would be appropriate to run regressions separately for each grassland, which would have increased the $r^2$ value, we instead more clearly described the relationship between simulated and observed SOC data. We do this in section 3.1. lines 604 – 636 where we elaborate on how we interpret the relationship between simulated and observed data. This is also mentioned in the discussion lines 768 – 778 where we broadly discuss model performance for the grasslands separately, and we acknowledge the model was less effective at simulating the acidic grassland.

*'Why do you choose to spin up the model for 10000 years, and how does the spin up time affect your results?'*

We hope this has been addressed in our previous comment relating to the spin up phase. In the aforementioned revised sections, we have included detail from our author response to this comment to explain why we use this spin up period, and how this period and the use of input driver data affect the simulation of contemporary C, N and P data.

*'A final comment, the discussion need to focus much less on the speculation of model outputs, but include some discussion about the possible caveats of model or study design and uncertainties caused by these limitations.'*

We agreed with this comment and as such have added a new section into the discussion that discusses the most relevant model limitations, but also provides some justification as to why N14CP is an appropriate model to use for this work. Discussion content that may be deemed speculative, or not relevant to model processes described in the methods section, has been removed.

This section is on lines 1052 – 1087 and includes:

- A summary of past model performance and some justification for our methodological approach.
- Discussion about the previously mentioned acidic C and P issue and what this tells us about the model and what this may mean for such P-limited systems.
- The model's simple plant nutrient pool structure and plant control over nutrient uptake (and potential ways to address this in future work).
- We also discuss some other limitations relating to biota-enhanced P weathering and nutrient redistribution.
- Finally, the importance of excluding the effects of elevated $CO_2$ on C, N and P dynamics is included at the end of this section.

**Overview of all revisions**

This section provides a textual summary of the changes to the revised manuscript, which are recorded in the tracked changes document but are provided here too for convenience.

**Title**

The title has been changed to better reflect the focus of the revised manuscript:

'Organic phosphorus cycling may control grassland responses to nitrogen deposition: a long-term field manipulation and modelling study'.

**Abstract**

Changes to the abstract reflect the clarification of the manuscript narrative and the refocussing of results.

- In line with recent developments in understanding of Wardlow, the calcareous grassland is now referred to as a limestone grassland, since this is a more precise definition and is in line with recent publications from the experimental site.
- The acidic grassland is no longer explicitly referred to as N-P co-limited.
    - The control 0N acidic grassland is likely to still be P-limited as it was at the onset of the experimental period.
    - Only after prolonged periods of N addition have we started to observe an N-limited response in LN and HN treatments, suggesting P limitation is starting to be alleviated
    - The acidic grassland is therefore likely not N-P co-limited, but has had P limitation lessened over time by N addition (consistent with the latest understanding of long-term N loading alleviating P limitation, Chen et al., 2020 Global Change Biology 26 5077-5086).
    - Accordingly, references to N-P co-limitation have been reduced throughout the revised manuscript.
- The results section of the abstract has been amended to highlight the importance of different levels of access to organic P in determining nutrient limitation and ecosystem responses to nutrient enrichment.
- Likewise, much of the original conclusions are the same but more emphasis is provided in the context of organic P cycling.

**1. Introduction**

- Lines 72 – 73
    - Fowler *et al.* (2013) reference updated to better contextualise anthropogenic impact on N cycling.
- Lines 92 – 94
    - Reference to Fay *et al.* (2015) study updated to be more concise.
- Lines 96 – 97
    - Findings by Du *et al.* (2020) made more relevant to P limitation as a whole rather than co-limitation by N and P.
- Lines 102 – 108
    - End of paragraph refocussed on P-limited rather than N-P co-limited ecosystems and some content has been cut.
- Lines 109 – 123
    - Explicit reference to the importance of plant and soil enzyme activity in P-limited grasslands, with discussion regarding the potential importance of phosphatase enzyme activity on organic P cleaving.
    - Contextualisation of the $P_{CleaveMax}$ parameter with empirical understanding of the Wardlow grasslands.
- Lines 124 – 129
    - The need to explore organic P cycling within N14CP, and its potential importance for understanding responses to changes in nutrient availability, is highlighted.
- Lines 138 – 139
    - We highlight that few modelling studies have explicitly investigated the role of organic P access in determining nutrient limitation.
- Lines 141 – 143
    - We comment on the value of exploring organic P acquisition within the model.
- Lines 147 – 151
    - Repetitive detail removed.
- Lines 152 – 156
    - Clarification that the two grasslands are P-limited and that they occur on contrasting soils, rather than they contrast in their limiting nutrient status.
- Lines 157 – 158
    - Brief statement of aim at the beginning of the paragraph.
- Lines 159 – 166
    - Removed unnecessary methodological detail.
- Lines 167 – 174
    - Aims section has been updated to better reflect the contents of the revised manuscript.
- Lines 175 – 182
    - Similarly, the hypotheses section has been revised to focus more on the potential roles of organic P access on P limitation and responses to anthropogenic nutrient input.

**2. Methods**

**2.1. Field experiment description**

- Lines 204 - 218
    - The description of the limiting nutrient status of the grasslands has been amended to highlight that both the acidic and limestone grasslands were initially P-limited but recently the N treated acidic plots have shown signs of alleviated P limitation.
    - Focus has therefore shifted to P-limited systems, with the contrasts between grasslands being related more to soil conditions than nutrient limitation.

**2.2. Summary of model processes**

2.2.1. N14CP model summary

- Lines 245 - 249
    - Highlighted the gap in our understanding of N14CP's ability to simulate ecosystems that are P-limited and when these are subjected to high levels of background N deposition.
    - Additional info on previous work looking at allowing $P_{Weath0}$ to vary to simulate contemporary C, N and P, which provides justification for us doing the same.
    - Highlighted that organic P cycling has not been explored using N14CP yet is likely to be important for ecosystems such as the Wardlow grasslands.
- Lines 250 - 253
    - Changes we have made to the model for the purpose of this study are described, in order to better distinguish what we do from the initial model development work.

2.2.2. Net primary productivity and nutrient limitations

- Line 257
    - Heading has been amended to include how we simulate nutrient limitation.
- Lines 258 - 259
    - Reference to spin up phase removed so it can be expanded upon in its own paragraph.
- Lines 260 - 262
    - We briefly mention how belowground biomass pools are represented.
- Lines 264 - 266
    - Introduction to nutrient limitation within N14CP, where we highlight that it is based on a Liebig's law of the minimum approach and what the factors limiting productivity are.
- Lines 267 – 280
    - Part of a detailed explanation of how nutrient limitation is determined within N14CP and how it relates to processes in the model that we calibrate to the site scale, such as plant functional type and its history.
    - By referring to the input driver section here, we hope that the role of such drivers in determining nutrient limitation are more clear.

- - o Changes in PFT stoichiometry in relation to nutrient perturbations are described, to give the reader more information about how litter quality may change in response to N enrichment, be it from experimental addition or deposition.
- Lines 281 – 284
  - o Removal of unnecessary text.
- Lines 287 - 293
  - o More detail on N14CP processes that should allow for nutrient co-limiting behaviour, despite simulating a single limiting nutrient.
  - o This allows our discussion of such processes to directly relate to model processes, as suggested by a reviewer.
  - o Information relating P availability to N fixation and its relationship with N deposition is provided.

2.2.3 Plant and soil N and P cycling

- Line 296
  - o Heading has been amended to accommodate the inclusion of more general information, not just of relevance to available N and P.
- Lines 297 - 300
  - o Inclusion of a short paragraph to direct the reader toward the new model summary figure (fig 1) and the original model development paper should they want to refer to it.
- Lines 301 - 307
  - o Plant available N and P paragraph has been revised to be more consistent with the new summary figure, and to make reference to the relationship between organic P cleaving and plant-available P.
- Lines 310 - 313
  - o More detail has been added to elaborate on what the contribution of Pweath0 is to different P pools, so its calibration is more relevant in the context of nutrient availability.
- Lines 317 – 333
  - o This whole section has been rewritten to more clearly explain the relationships between nutrient limitation, P availability and organic P cleaving.
  - o The first paragraph in this section describes how the size of the available P pool is calculated, and from which sources, in addition to how the model accesses each of these compartments.
  - o $P_{Cleave}$ is introduced, prior to discussing our modification of the $P_{CleaveMax}$ parameter so the reader can see how this has developed in this work from the initial model development paper.
  - o As per a reviewer comment, we have included a rate equation from the initial N14CP paper (Equation 1) to quantify this parameter and how it is limited to 'realistic' quantities given the lack of empirical data with which to parameterise it.
  - o Finally, we include more detail about what we do in this study, including the reference to $P_{CleaveMax}$, what it is and how it differs from $P_{Cleave.}$
- Lines 334 – 367
  - o Removed / rearranged text.
- Lines 368 - 378

- More detail about the simulation of SOP alongside other SOM constituents and its subsequent inclusion into available P pools.
- C, N and P loss processes have been moved to the end of this section to make the structure clearer.

**Figure 1**

- Figure 1 and its legend has been completely revised to include more of the initial model processes, to make it clearer and to highlight what has been changed in the current study compared to the initial model development paper.
- The figure legend has also been substantially revised to highlight changes for the purpose of this work, to describe the key relating to the new processes, pools and fluxes and to represent some of the nutrient inter-dependencies we mention in the discussion.
- The schematic now represents C, N and P flows across all different ecosystem compartments of the model (atmosphere, plants, topsoil, subsoil/leachate) and not just contributions to available N and P pools.

- Lines 391 – 422
    - Removed / rearranged text.

**2.3. Simulating the field experiment with the model**

- Lines 425 – 437
    - A summary paragraph describing how we use the empirical data to achieve the aims of the work.
    - A brief reminder of the aims is included at the end of this paragraph to make the following section's relevance more clear.

2.3.2 Input drivers

- Lines 449 – 464
    - More detail is provided regarding the use of input driver data and justification for the model spin up phase.
    - Justification for spinning up from the onset of the Holocene, including information on why we do this and clarification that this isn't an attempt to simulate the whole spin up phase, but rather ensure that our calibration takes into account past conditions and forcings.
    - To address reviewer concerns about how we can use such a spin up period to simulate contemporary soil stocks, we provide more information about the role of site-specific input drivers of varying temporal scales.
    - We justify the spin up approach for ecosystems not likely to be in equilibrium, such as those exposed to high levels of N deposition.
    - More information about how we simulate the PFT history to the site scale.

- Lines 465 – 470
    - Description of the PFT history of the modelled grasslands is removed as it was deemed of little relevance.
    - This was moved and amended from lines 488 – 495 prior to removal, which is why it appears twice.

2.3.3. Model parameters for the acidic and limestone grasslands

- Lines 513 – 523
    - Removed this paragraph as it repeats much of what is said earlier in the methods and intro.
- Lines 524 - 528
    - More justification for exploring the $P_{CleaveMax}$ parameter within the model, and we highlight that empirical quantification of this parameter is poor.
- Lines 537 and 541
    - Amended mistakes where we suggested that aboveground biomass C was included in the cost function when it is not.
- Lines 551 – 565
    - Extensive detail to justify our methodological approach and clarify our selection of model parameters.
    - Explanation that plant biomass C was excluded from the cost function and justification as to why this was deemed appropriate.
    - In addition, we justify why we included data from all nutrient treatments and kept a variable separate for testing.
    - We briefly describe the limitations of the cost function approach, and highlight that this only allows us to capture part of the empirical responses to nutrient treatment
    - We then explain why the pattern of responses, i.e. how each variable responds to treatment, is important and how we account for this.
    - This highlights the relevance of the regression statistics used and described in the results section regarding figure 2.

3. Results

- Lines 571 – 579
    - An introductory paragraph to relate the following results and figures to the aims of the work and the preceding methodology section.
    - This section highlights the purpose of each figure.

3.1. Varying phosphorus source parameters

- Lines 583 – 603
    - Revised / rearranged text.
- Lines 604 – 636

- o This section has been rewritten to better reflect the purpose of using N14CP to simulate Wardlow and to more clearly describe the relationships between empirical and modelled data, including more explicit references to the $r^2$ and pattern of responses.
- o This section now contains:
- o A paragraph detailing what the selected combinations of $P_{CleaveMax}$ and $P_{Watch0}$ were.
- o Brief overview paragraph of what Figure 2 shows and overall performance, with some information about the model's simulation of the most limiting nutrient for each grassland.
- o Some data on overall performance (across all variables) of the acidic and limestone grasslands separately, with reference to magnitude and pattern of data.
- o We note the caveat of low $r^2$ values for some variables but demonstrate that the $r^2$ of the testing data, aboveground biomass C, is large and is consistent with the empirical data.
- o A section for describing SOC performance, including acknowledgement of the low $r^2$ value and the lack of an increase in SOC P addition in the acidic grassland.
- o Section describing model performance at simulating magnitude and pattern of empirical SON data.
- o Section describing model P performance, same as above SOC and SON sections.
- o Caveat that the $r^2$ value is of little relevance for so few data points.

**Figure 2**

- Panels b) SOC and c) SON for figure 2 have had their axes shortened to spread the data points to make them easier to distinguish as per reviewer comments.
- The legend has been updated to correct the final sentence where we imply that the $r^2$ value is used to assess closeness to the 1:1 line – this is inaccurate and has been amended to more accurately reflect what this value shows (direction of response of each variable to nutrient addition).
- We also include a caveat at the end to highlight that a low $r^2$ value is not necessarily indicative of poor model fit.
- The colour scheme has been amended to be more colour-blind friendly, swapping green for cyan to aid those that cannot distinguish between red and green.
- The figure legend now refers to limestone rather than calcareous.

**3.2. The limiting nutrient through time**

- Only minor changes were made to this section.

**3.3. Modelled trends and responses to nutrient additions**

- Lines 672 – 673
  - o Edit to clarify that the timeseries runs from the onset of large-scale N deposition defined as starting in 1800 within the model.

- Other minor changes made to this section's text include replacing calcareous with limestone and increasing readability more generally.

**Figure 4**

- A useful suggestion from a reviewer was to include the empirical data from figure 2 into this figure to better contextualise the timeseries data – these have been added, and both legends (text legend and figure legend) have been amended to account for this inclusion.
- We shortened the x axis by 50 years to show data from 1850 instead of 1800, and the axes have been extended beyond 2020 so the experimental period is clearer.
  - While it was suggested to reduce the x axis further than this, we feel providing this timescale more clearly demonstrates the magnitude of nutrient manipulation compared to background effects of N deposition, and shows how these pools developed through time.
- The inset subplots have had their x axis values removed so they fit better above the timeseries lines.
- As with figure 2, the colour scheme has been amended to be more colour blind friendly and calcareous has been replaced with limestone.

**4. Discussion**

The discussion has been re-written to account for the updated narrative of the manuscript – as such, tracked changes appear to replace the entire section, though much of the old detail is still there, it is just structured to be more clear and made more concise.  The main changes are as follows:

**4.1. Simulating contrasting grasslands by varying access to P sources**

- The previous summary section has been removed, and some detail from it has been incorporated into the opening paragraph of this section.
- The subheading title has also been changed to better reflect the contents of this section.

- Lines 749 – 757
  - A brief reminder of how we used empirical data to model the grasslands, and how we used the simulated grasslands to achieve our initial aims.
  - This allows the results section to flow more smoothly into the discussion rather than the previous MS version where the main results were just repeated as the first paragraph.
  - We also re-emphasise that the objective of the modelling was not to perfectly reproduce Wardlow but rather to test how varying access to different P sources within the model allows us to develop our understanding of the model's representation of P cycling, and to contextualise these results with empirical data
- Lines 758 – 767

- o  Elaboration on what the selected parameter combinations mean for the grasslands and how that may reflect the topography of the different Wardlow grasslands.
  - o  This paragraph highlights how the potential difference in relative P availability of the empirical grasslands was captured by the model, where the acidic had higher access to organic P compared to the limestone, which had greater inorganic P availability.
- Lines 768 – 778
  - o  Summary of the model's overall performance in simulating the two grasslands
  - o  Includes a summary statistic of average discrepancy between empirical and modelled data and how effectively the model captured the nutrient-limiting response of both grasslands.
  - o  We explicitly highlight the model's inability to simulate a positive response to P addition in the acidic grassland, despite the empirical data showing such a response, and set up the next paragraph to discuss why.
- Lines 779 – 788
  - o  We acknowledge here that the underestimation in soil P and an overestimation of plant and soil C in the acidic grassland likely indicates the model is allowing too much P to leave the soil and stimulate plant growth.
  - o  We suggest that despite the associated difficulties, distinguishing between inorganic and organic P forms in the empirical data we used for calibration would have been helpful.
- Lines 789 – 798
  - o  To address reviewer concerns, this paragraph explains why we believe the acidic C data is overestimated and the P data underestimated.
  - o  We explain how the model's use of the $P_{CleaveMax}$ parameter likely underestimates the ability of soil surfaces and microorganisms to immobilise available P in solution.
  - o  Plant uptake of newly-cleaved P is therefore likely too high in the simulated acidic grassland, simultaneously leading to underestimated total P (as P is removed from soil and transferred to biomass) and an increase in biomass and soil C, as stimulated plant growth increases carbon accumulation.
  - o  We add additional detail from a reference suggested by a reviewer to put this more into context of phosphatase enzymes, to show how our conclusion is consistent with empirical understanding of organic P cycling.
  - o  In addition, we suggest that improved modelled access to secondary forms of inorganic mineral P, such as iron, aluminium and calcium phosphate complexes, may help us better understand the differences in P cycling between the empirical grasslands.
- Lines 799 – 813
  - o  Discussion of the model's representation of N-P co-limited behaviour and how such representations may account for the lack of response to P addition in the acidic grassland.
  - o  More detail about the relationship between N fixation and P availability is provided, building on the relevant additions to the methods section.
  - o  How we believe this mechanism is affected by site-specific levels of N deposition and the consequences of such an effect on model behaviour are also described.
  - o  This paragraph has been moved to the end of this section so more emphasis is on the organic P cycling discussion.
- Lines 814 – 816

- o A sentence has been added to briefly summarise this section, so it doesn't abruptly end like it previously did.

**4.2. Consequences of differential P access on ecosystem C, N and P**

- Subheading title has been revised to better reflect contents of this section and the main aims of the study.

- Lines 822 – 828
  - o A brief introductory paragraph to this section to more directly relate the model's estimation of P access conditions to the timeseries results and simulating the experimental period, and the contrasting results of the two modelled grasslands.
- Lines 829 - 840
  - o More detail about what the different $P_{CleaveMax}$ values mean for the two grasslands, and how the SOP timeseries data suggests they may be responding to experimental nutrient manipulation.
  - o Here we explain more clearly how P dynamics differ between the two simulated grasslands, and highlight that the SOP data suggests N treatment increased plant P demand in both.
- Lines 841 - 848
  - o Here we relate the high access of organic P in the acidic grassland to the model simulating it as N limited.
  - o We explain the acidic SOC data in the context of SOP access and the most limiting nutrient predicted by the model.
  - o We believe this more coherently links different aspects of the work (simulating SOP access, nutrient limitation and SOC dynamics) than it did before.
- Lines 849 – 854
  - o This brief paragraph contextualises the modelling results with empirical studies showing similar conclusions of positive SOC responses to N addition.
- Lines 855 – 866
  - o Discussion about the acidic grassland empirical phosphatase activity and its similarities to the modelled grassland – references the Chen *et al.* 2020 hypothesis whereby long term N deposition alleviated P limitation by stimulating phosphatase enzymes for a sufficient time period.
  - o We suggest that such high rates of SOP access are unlikely to occur outside of such high N loading scenarios as with our LN and HN treatments.
- Lines 867 – 878
  - o The response of the limestone grassland plant and soil C to nutrient manipulation, and an explanation for why that happens within the model.
  - o We suggest that if similar processes were to occur in empirical systems, ecosystem SOC stocks may reduce
- Lines 880 – 886
  - o Removed references to empirical studies showing contrasting results of N and P enrichment on SOC stocks, as we believe it added too much focus on this particular result without providing sufficient contextualisation for it.
- Lines 887 – 895

- o Removed section describing empirical responses of C efflux to N deposition, as these were not directly related to model processes described in the methodology section.
- Lines 896 – 1050
  - o A large section of the old discussion, much of which has been removed, refocussed or rearranged to be more relevant to the revised discussion.

**4.3. Model limitations**

- This new section (lines 1052 – 1087) describes some of the limitations of the model and caveats to the study's findings.
- Lines 1054 – 1060
  - o Brief discussion of how N14CP has been used in the past to justify its use here, and an identification for the need to explore organic P cycling within the model.
- Lines 1061 – 1070
  - o Discussion of N14CP's simplified pool structure and the limited control of nutrient uptake by plants.
  - o We discuss how our approach of varying $P_{CleaveMax}$ to an extent aimed to address this.
  - o We discuss caveats of our methodological approach.
- Lines 1071 - 1080
  - o We acknowledge that the inclusion of an inorganic P equivalent to organic cleaving to represent other forms of plant nutrient acquisition may be a useful addition to the model.
  - o We discuss other potential limitations of N14CP, such as the potential effects of biota-enhanced P weathering and nutrient redistribution and why these are not included within N14CP.
- Lines 1081 - 1087
  - o Limitation of not including elevated $CO_2$ in the model, and what impact this may have for nutrient cycling, and how C availability is currently determined.

**5. Conclusions**

- Lines 1090 – 1096
  - o We highlight how varying P acquisition within N14CP has allowed us to account for contrasting responses of two P-limited grasslands.
  - o Here we acknowledge that such coarse representation of organic P access within the model likely limits our ability to accurately simulate grasslands where N and P interact to control plant productivity, including the potential for N inputs to alleviate P limitation.
- Lines 1097 – 1111
  - o Much of this section has been removed to be more consistent with the overall narrative of the paper.
  - o We highlight how the model simulated an N limited and P limited grassland from the data we provided it, and that access to organic P played an important role in this.
  - o We summarise the key differences between the responses of the modelled grasslands and explain how lack of access to organic P reduced SOC stocks in the modelled limestone grassland.

- Lines 1112 – 1115
  - A brief paragraph to describe the effects of modelled background N deposition on the formation of ecosystem C, N and P pools.
- Lines 1116 – 1123
  - The previous N deposition section has been reduced to focus conclusions more on organic P cycling results.
- Lines 1123 - 1128
  - A final paragraph to summarise the potential implications of our findings.
  - Access to sufficient organic P may allow ecosystems to respond positively to N addition, whereas insufficient access has the opposite effect, eventually reducing SOC stocks.

**References**

- References have been updated accordingly.

---

## Referee Report (RR1)

I carefully read the manuscript as well as the authors' reply to the previous round of referee comments. From my point of view, the authors have done a good job addressing the reviewer comments. The limitations are now better discussed.  I just have some additional suggestions

- The manuscript is generally very lengthy
- Abstract sentences 1 and 2 are contradicting. Suggest to delete the first sentence
- Above ground biomass carbon, soil organic carbon, and total N were newly measured for this study. I was confused that the methods were not described and the results not shown in the main text, and only later found a description of the methods in the supplement. However, soil analyses are not defined appropriately. They are at once way too lengthy and on the other hand lacking any citation. Such basic measurements as above ground biomass, soil organic carbon and total N should be done according to standard protocols. This should be cited appropriately. I would suggest to move the methods section (in a more concise) form to the main text. Also, I would suggest to start the results section with these observational results (leaving out the modelling in a first table or figure). This will help the reader get to know these sites and how they respond to the different treatment, which are then the basis for interpreting the model.
- l. 223-234 P deposition is assumed to be negligible in this model. Actually, more and more evidence is showing that P deposition is just as important as rock weathering for P inputs to terrestrial ecosystems (see e.g. Aciego et al. 2017). This should be considered for further model development in the future
- to improve readability, I suggest to reduce abbreviations. Specifically, no need to abbreviate PFT
- l. 349, replace the coma with a period
- l. 628 authors state organic P release from SOM and immobilization are poorly represented in models and that they encourage further study to quantify these processes. I agree with these statements; however, from reading the manuscript I wondered if the authors were aware of the state of the art P flux measurements since the results are not discussed in light of measurement data? Several studies have actually measured organic P mineralization and microbial immobilization with radioisotopes, and would be relevant for interpreting the modelling results presented here. For example, Bünemann et al. 2012 looked at mineralization fluxes in grasslands under NPK treatments and Schneider et al. 2017 calculated organic P fluxes in calcerous soil.
- General remark on over-selling: authors should be careful not to overinterpret their results stemming from modelling two grassland soils, especially given the limitations as discussed.
- L. 46 Wardlow is not a globally important C sink. Please delete this sentence, since it is not appropriate to extrapolate from two sites simulated here onto a global level
- L. 641 same here. It is inappropriate to generalize from the two grassland sites about ecosystems in general all over the world.
- L. 648 I don't consider N14CP to be "one of the first" CNP models. Many other models come to mind, some of which much older or much more developed: JSBACH, CABLE-CNP, CLM-CNP, ORCHIDEE-CNP, QUINCY, ForSAFE, …

References

Aciego, S., Riebe, C., Hart, S. *et al.* Dust outpaces bedrock in nutrient supply to montane forest ecosystems. *Nat Commun* **8,** 14800 (2017). https://doi.org/10.1038/ncomms14800

Kimberley D. Schneider, R. Paul Voroney, Derek H. Lynch, Astrid Oberson, Emmanuel Frossard, Else K. Bünemann, Microbially-mediated P fluxes in calcareous soils as a function of water-extractable phosphate, Soil Biology and Biochemistry, Volume 106, 2017, Pages 51-60, https://doi.org/10.1016/j.soilbio.2016.12.016.

E.K. Bünemann, A. Oberson, F. Liebisch, F. Keller, K.E. Annaheim, O. Huguenin-Elie, E. Frossard, Rapid microbial phosphorus immobilization dominates gross phosphorus fluxes in a grassland soil with low inorganic phosphorus availability,
Soil Biology and Biochemistry, Volume 51,
2012, Pages 84-95, https://doi.org/10.1016/j.soilbio.2012.04.012.

---

## Author Response (AR2)

We would like to thank both reviewers for taking the time to review our work and for providing constructive comments to improve the manuscript. Please find our responses to reviewer comments below in italics, with the original comments in regular text.

Note that our references to line numbers refer to those of the tracked changes document.

**Reviewer 1**

'Line 276: I'm not sure why the flexible stoichiometry here is described as representing changes in species only rather than ecosystem level changes which can be either plastic changes or shifts in species compositions, since the model (as most other models) cannot differentiate between the two?'

*The text here has been amended to reflect that the changes in stoichiometry do not just occur at the species level and extend to the ecosystem scale as the reviewer points out. The updated paragraph is on lines 201 – 206:*

*'Stoichiometry of coarse tissue is constant but the fine tissue of each plant functional type has two stoichiometric end members. This allows the model to represent transitions from N-poor to N-rich plant communities or an enrichment of the fine tissues within plants (or a combination of both) [Davies et al., 2016b], dependent on available N. '*

'Line 290: Is BNF regulated by anthropogenic deposition only or more generally by soil N content? This might help the reader understand the model-data discrepancy under P addition in the acidic grassland.'

*BNF in the model is downregulated by N deposition only, not soil N content more generally as it is assumed that atmospherically deposited N is readily available to N fixers, whereas soil N is not necessarily available. We have clarified this in the methods section lines 212 – 217:*

*'The initial rate of N fixation is based on literature values for a given plant functional type and is downregulated by anthropogenic N deposition, but not soil N content more generally, as it is assumed that atmospherically deposited N is readily available to N-fixers. Nitrogen fixation in the model is also related to P availability. The degree to which P availability limits this maximum rate of fixation is determined by a constant; $K_{Nfix}$ [Davies et al. 2016b]'*

'Line 301: Why coarse litter and not all litter?'

*Within the model, coarse litter forms the litter pool and decomposes on the soil surface to contribute N and P to the bioavailable pool. Fine litter is not included in this pool and due to its more rapid turnover time, it is incorporated directly into the SOM pool, where it can then decompose to contribute nutrients to the bioavailable N and P pool. A sentence to clarify this has been added on lines 228 – 231:*

*'Plant available N is derived from biological fixation, the decomposition of coarse litter and SOM, atmospheric deposition and direct N application. Fine plant litter enters the SOM pool directly due to its rapid rate of turnover whereas coarse litter contributes N and P through decomposition and does not join the SOM pool.'*

'Line 606: Onwards check the precision of your reported values, there are some inconsistencies'

*Thank you for pointing this out, we have amended the precision of the reported $P_{CleaveMax}$ values to two decimal places so they are consistent with the reporting of the $R^2$ values later in the text. These values are on lines 416 and 417. The reported precision for the percentage difference between empirical and modelled data and their standard errors remain to one decimal place, which we felt was more appropriate for the magnitude of the data.*

**Reviewer 2**

'The manuscript is generally very lengthy'

*We agree with the reviewer that the manuscript is fairly long, though its length is comparable to other Biogeosciences manuscripts, particularly those detailing modelling results and methodology. Through our revisions we have cut much content, particularly from the discussion, though this has largely been replaced by additional text in the methodology to more clearly describe the model as requested by the reviewers, to ensure the reader can effectively interpret our results.*

'Abstract sentences 1 and 2 are contradicting. Suggest to delete the first sentence'

*The first sentence of the abstract has been revised so that it is no longer contradictory and to better integrate the theme of P limitation in C, N and P cycling. The new sentence on lines 29 – 31 now reads:*

*'Phosphorus (P) limited ecosystems are widespread, yet there is limited understanding of how these ecosystems may respond to anthropogenic deposition of nitrogen (N), and the interconnected effects on the biogeochemical cycling of carbon (C), N and P.'*

'Above ground biomass carbon, soil organic carbon, and total N were newly measured for this study. I was confused that the methods were not described and the results not shown in the main text, and only later found a description of the methods in the supplement. However, soil analyses are not defined appropriately. They are at once way too lengthy and on the other hand lacking any citation. Such basic measurements as above ground biomass, soil organic carbon and total N should be done according to standard protocols. This should be cited appropriately. I would suggest to move the methods section (in a more concise) form to the main text. Also, I would suggest to start the results section with these observational results (leaving out the modelling in a first table or figure). This will help the reader get to know these sites and how they respond to the different treatment, which are then the basis for interpreting the model.'

*Thank you for these suggestions, we have implemented most of the recommended revisions. We have cut unnecessary content from the description of the empirical methods, and provided relevant citations for standard protocols, and detailed if and how these were adapted. In addition, and as proposed, we have put all empirical data into a table separate from any modelled data, to aid the reader in interpreting empirical responses to nutrient treatments. This also necessitated the removal of some redundant information from later data tables.*

*However, we believe that keeping these empirical methods and data in the supplementary material is preferable, given our intended focus of the manuscript. Firstly, and as identified by the reviewer, the manuscript is already fairly long, hence we would be hesitant to add detail to the methods, results and potentially discussion section when all the information is accessible in the supplementary material (in a now more succinct format). We have added a couple of sentences in the methods (lines 143 - 144) and results (lines 403 – 404) to direct readers who are particularly interested in the empirical data.*

*We have discussed including the empirical data in the main manuscript and we feel that its inclusion may detract from what is at its core a modelling paper, and perhaps lead to interpretations that we did not intend. For instance, the narrative has been revised in the current version to clarify that we were not attempting to replicate the empirical data and instead used it to inform model development and interpret our model outputs. Adding the empirical data back into the main text may be counter-productive to this aim. Instead, we opted to describe the empirical results in context of the modelling in a more qualitative way and provide the empirical data in the supplementary information for interested readers.*

'Lines 233 – 234: P deposition is assumed to be negligible in this model. Actually, more and more evidence is showing that P deposition is just as important as rock weathering for P inputs to terrestrial ecosystems (see e.g. Aciego et al. 2017). This should be considered for further model development in the future'

*Thank you for highlighting this interesting study. We assume that P deposition is negligible here as we cannot account for losses of P that also occur through landscape redistribution, which is an issue discussed in a meta-analysis some co-authors were involved with (Tipping et al. 2014). We do agree though, that in combination with improved empirical data on P deposition and redistribution, this would make an interesting modelling study for the future. We have elaborated on our assumption in the methods lines 242 – 246:*

*'In principle, P can be added in small quantities by atmospheric deposition [Ridame and Guieu, 2002] but for the purpose of this work, P deposition is set to zero in the model. While the contribution of P through atmospheric deposition is increasingly realised [Aciego et al. 2017], we cannot account for the losses of P that may also occur through landscape redistribution [Tipping et al. 2014].'*

'To improve readability, I suggest to reduce abbreviations. Specifically, no need to abbreviate PFT'

*Plant functional type is no longer abbreviated throughout the text.*

'Line 349, replace comma with a period'

*Amended.*

'Line 628: Authors state organic P release from SOM and immobilization are poorly represented in models and that they encourage further study to quantify these processes. I agree with these statements; however, from reading the manuscript I wondered if the authors were aware of the state of the art P flux measurements since the results are not discussed in light of measurement data? Several studies have actually measured organic P mineralization and microbial immobilization

with radioisotopes, and would be relevant for interpreting the modelling results presented here. For example, Bünemann et al. 2012 looked at mineralization fluxes in grasslands under NPK treatments and Schneider et al. 2017 calculated organic P fluxes in calcareous soil'

*Thank you for these suggestions. We were not aware of these measurements and have incorporated some of their findings into the discussion. In particular, we have added a brief paragraph to contextualise some of our modelling results with the findings of the recommended papers (lines 578 – 585):*

*'In addition to physico-chemical processes reducing P availability, in P-limited grassland soils, microbial processes may be dominant drivers of ecosystem P fluxes [Bünemann et al. 2012]. For instance, while mineralisation of organic P may increase inorganic P in solution [Schneider et al. 2017], this can be rapidly and almost completely immobilised by microbes, particularly when soil P availability is low [Bünemann et al. 2012]. As the model lacks a mechanism for increasing access to secondary mineral P forms comparable to organic P-cleaving, and microbial P immobilisation is incompletely represented for P-limited conditions, it is possible that the uptake of organic P by the acidic grassland in the model is exaggerated.'*

'General remark on over-selling: authors should be careful not to over interpret their results stemming from modelling two grassland soils, especially given the limitations as discussed.'

*We have amended our interpretation of the results so as not to extend them beyond what is reasonable (see related comments below for specifics).*

'Line 46: Wardlow is not a globally important C sink. Please delete this sentence, since it is not appropriate to extrapolate from two sites simulated here onto a global level'

*The concluding sentence of the abstract (lines 46 – 48) has been revised so it no longer implies Wardlow is a globally important carbon sink. This was not our initial intention but the wording was problematic as the reviewer identifies. The sentence now reads:*

*'We conclude that grasslands differing in their access to organic P may respond to N deposition in contrasting ways and where access is limited, soil organic carbon stocks could decline.'*

'Line 641: Same here. It is inappropriate to generalize from the two grassland sites about ecosystems in general all over the world.'

*We have changed lines 664 – 667 so they do not generalise to ecosystems around the globe and instead make explicit reference to ecosystems such as Wardlow:*

*'This suggests that in P-limited limestone grasslands such as at Wardlow, where access to organic P forms may be comparatively limited, N deposition may worsen pre-existing P limitation and reduce ecosystem C stocks [Goll et al. 2012, Li et al. 2018].'*

'Line 648: I don't consider N14CP to be "one of the first" CNP models. Many other models come to mind, some of which much older or much more developed: JSBACH, CABLE-CNP, CLM-CNP, ORCHIDEE-CNP, QUINCY, ForSAFE'

*This is a fair comment and we have amended the text accordingly on lines 673 – 675. We highlight that N14CP is one of few but not one of the first:*

*'While N14CP is a fairly simple ecosystem model by design, it is one of few models to integrate the C, N and P cycles for semi-natural ecosystems and has been extensively tested against empirical NPP and soil C, N and P data.'*